# Properties and dynamics of meron topological spin textures in the two-dimensional magnet CrCl$_3$

Mathias Augustin[1], Sarah Jenkins[2], Richard F. L. Evans [2], Kostya S. Novoselov[3,4] & Elton J. G. Santos [5,6✉]

Merons are nontrivial topological spin textures highly relevant for many phenomena in solid state physics. Despite their importance, direct observation of such vortex quasiparticles is scarce and has been limited to a few complex materials. Here, we show the emergence of merons and antimerons in recently discovered two-dimensional (2D) CrCl$_3$ at zero magnetic field. We show their entire evolution from pair creation, their diffusion over metastable domain walls, and collision leading to large magnetic monodomains. Both quasiparticles are stabilized spontaneously during cooling at regions where in-plane magnetic frustration takes place. Their dynamics is determined by the interplay between the strong in-plane dipolar interactions and the weak out-of-plane magnetic anisotropy stabilising a vortex core within a radius of 8–10 nm. Our results push the boundary to what is currently known about nontrivial spin structures in 2D magnets and open exciting opportunities to control magnetic domains via topological quasiparticles.

[1] School of Mathematics and Physics, Queen's University, Belfast BT7 1NN, UK. [2] Department of Physics, The University of York, York YO10 5DD, UK. [3] Department of Material Science & Engineering, National University of Singapore, Block EA, 9 Engineering Drive 1, Singapore 117575, Singapore. [4] Chongqing 2D Materials Institute, Liangjiang New Area, 400714 Chongqing, China. [5] Institute for Condensed Matter Physics and Complex Systems, School of Physics and Astronomy, The University of Edinburgh, Edinburgh EH9 3FD, UK. [6] Higgs Centre for Theoretical Physics, The University of Edinburgh, Edinburgh EH9 3FD, UK. ✉email: esantos@ed.ac.uk

The finding of magnetism in atomically thin vdW materials has attracted much recent interest[1–4]. The strict 2D nature of the layers leads to unique physical properties ranging from stacking dependent interlayer magnetism[5,6], giant tunneling magnetoresistance[7,8], and second harmonic generation[9], up to electric field control of magnetic properties[10]. Of particular interest are topological spin excitations[11,12], e.g., merons, which are crucial to understand the fundamental problems of chiral magnetic order[13] and the development of novel spintronic devices for information technologies[11]. Layered magnetic materials provide an ideal platform to investigate and harness this critical spin phenomenon as the genuine character of meron systems is intrinsically 2D and the integration of magnetic sheets in device heterostructures is a reality[14].

Here we demonstrate that monolayer CrCl₃ hosts merons and antimerons in its magnetic structure. Both quasiparticles are created naturally during zero-field cooling at low temperatures. We find that both spin textures are directly associated with the metastability of the magnetic domains on CrCl₃ induced by spin fluctuations. The merons and antimerons assume a random distribution throughout the surface creating a network of topological spin textures with no apparent lattice-order as observed in other materials[13]. The different sites of the network can interact with each other leading to a different types of collisions involving meron and antimerons occurring within a nanosecond time scale. Our results indicate that the control of vortex and antivortex in CrCl₃ is also a driving force for the manipulation of magnetic domains, which follows closely the annihilation process of the spin textures.

## Results

Our starting point is the following spin Hamiltonian:

$$\mathcal{H} = -\sum_{ij} J_{ij}(\mathbf{S}_i \cdot \mathbf{S}_j) - \sum_{ij} \lambda_{ij} S_i^z S_j^z - \sum_i D_i(\mathbf{S}_i \cdot \mathbf{e}_i)^2$$
$$- \sum_{ij} K_{ij}\left(\mathbf{S}_i \cdot \mathbf{S}_j\right)^2 - \sum_i \mu_i \mathbf{S}_i \cdot \left(\mathbf{B}_i + \mathbf{B}_i^{dp}\right), \tag{1}$$

where $\mathbf{S}_i$ and $\mathbf{S}_j$ are the localized magnetic moments on Cr atomic sites $i$ and $j$, which are coupled by pairwise exchange interactions. $J_{ij}$ and $\lambda_{ij} = J_l^{xx,yy} - J_l^{zz}$ (where $l$ sets the nearest neighbors taken into account) are the isotropic and anisotropic bilinear (BL) exchanges, respectively, and $D_i$ is the single-ion magnetic anisotropy. $\mathbf{B}_i$ and $\mathbf{B}_i^{dp}$ represent external and dipole magnetic field sources respectively. We considered in Eq. (1) up to third-nearest neighbors for $J_{ij}$ ($J_1 - J_2 - J_3$) and $\lambda_{ij}$ ($\lambda_1 - \lambda_2 - \lambda_3$) in the description of CrCl₃. The fourth term in Eq. (1) represents the biquadratic (BQ) exchange, which involves the hopping of two or more electrons between two adjacent sites[15,16]. Its strength is given by the constant $K_{ij}$, which is the simplest and most natural form of non-Heisenberg coupling. We recently found that several 2D magnets develop substantial BQ exchange in their magnetic properties, which is critical to quantitatively describe important features such as Curie temperatures, thermal stability, and magnon spectra[16]. The magnitude of $K_{ij}$ for CrCl₃ is 0.22 meV, which is slightly smaller than the BL exchange for the first-nearest neighbors $J_1 = 1.28$ meV but too large to be ignored. In our implementation, the BQ exchange is quite general and can be applied to any pairwise exchange interaction of arbitrary range. All the parameters in Eq. (1) are extracted from non-collinear ab initio simulations including spin–orbit coupling to determine the different components of $J_{ij}$, $\lambda_{ij}$, and $K_{ij}$ ($ij = xx, yy, zz$) as described in ref. [16]. For atomistic spin dynamics, we calculate the effective magnetic field $B^i$ arising from the BQ exchange

interactions with components:

$$B_x^i = -\frac{1}{\mu_i}\frac{\partial \mathcal{H}}{\partial S_x} = 2K_{ij}S_x^j\left(S_x^i S_x^j + S_y^i S_y^j + S_z^i S_z^j\right)$$
$$B_y^i = -\frac{1}{\mu_i}\frac{\partial \mathcal{H}}{\partial S_y} = 2K_{ij}S_y^j\left(S_x^i S_x^j + S_y^i S_y^j + S_z^i S_z^j\right) \tag{2}$$
$$B_z^i = -\frac{1}{\mu_i}\frac{\partial \mathcal{H}}{\partial S_z} = 2K_{ij}S_z^j\left(S_x^i S_x^j + S_y^i S_y^j + S_z^i S_z^j\right).$$

This effective field is then included within the total field $\mathbf{B}_{eff}$ describing the time evolution of each atomic spin using the stochastic Landau-Lifshitz-Gilbert (LLG) equation:

$$\frac{\partial \mathbf{S}_i}{\partial t} = -\frac{\gamma}{(1+\lambda^2)}[\mathbf{S}_i \times \mathbf{B}_{eff}^i + \lambda \mathbf{S}_i \times (\mathbf{S}_i \times \mathbf{B}_{eff}^i)], \tag{3}$$

where $\gamma$ is the gyromagnetic ratio. See Supplementary Section 1 for details. The dynamics of merons and antimerons is determined at different temperatures $T$ and magnetic fields $\mathbf{B}_{eff}$ over a time scale of more than 40 ns since the cooling starts at $T \gg T_c$, where $T_c$ is the Curie temperature, until it reaches 0 K within 2 ns (Supplementary Section 2). Figure 1 shows that as this process occurs, the nucleation of several small areas with an out-of-plane spin polarization $S^z$ perpendicular to the easy-plane of CrCl₃ appeared naturally at $T < T_c$ (Supplementary Fig. S1 and Supplementary Movie S1). As the temperature reached 0 K, there is a clear formation of several notch structures that are created randomly all over the crystal without following any apparent pattern or preferential nucleation site (Fig. 1c). Indeed, the number of dark and bright spin textures formed during the cooling at zero field is even indicating some equilibrium on the different signs of $S^z$. A similar nucleation process also happened at a finite magnetic field perpendicular to the surface within the range of 5–10 mT (Supplementary Fig. S2). Major modifications are observed at a magnitude of 50 mT as the field breaks the reflection symmetry of the layer, thereby leading to an asymmetry with respect of the polarization of the spin textures created (Fig. 1d–f and Supplementary Movie S2). That is, more dark spin complexes are nucleated, which follow the direction of the out-of-plane external field. The darker background relative to the zero-field situation indicates that the spins gained an additional $S^z$ component. At fields close to 100 mT just one kind of spin polarization is observed throughout the spin textures, which vanished completely for larger magnitudes (Supplementary Fig. S3 and Supplementary Movie S3). We performed a partial hysteresis calculation to estimate the critical magnetic field to switch all the spins in the system including those at the distinct textures (Supplementary Fig. S4). We found a magnitude of 200 mT, which is surprisingly large for the small single-ion anisotropy but consistent with recent magnetometry measurements[17].

In order to identify the nature of the dark and bright spin structures, we analyzed closely the different patterns formed on CrCl₃ at zero field and low temperatures (Fig. 2). We could identify four main spin textures by looking at the spatial distribution of $S^z$ as labeled B1–B4 (Fig. 2a). Even though they show similar magnitudes of $S^z$ along $x$ or $y$ (Fig. 2b) the same does not apply for components $S^x$ and $S^y$. That is, some shape anisotropy is observed at the in-plane distribution of the magnetization. We estimated an average radius of around 14.5 and 15.7 nm along of $x$- and $y$-displacement, respectively, despite of the spin notches considered (Supplementary Section 3). This indicates a more elliptical pattern where the main distinction between the spin textures B1–B4 is the orientation of the spins from the core (out-of-plane) to in-plane away from the center forming a vortex structure. Such vortex structures are typical of non-trivial topological spin textures such as skyrmions and antiskyrmions[18]. To

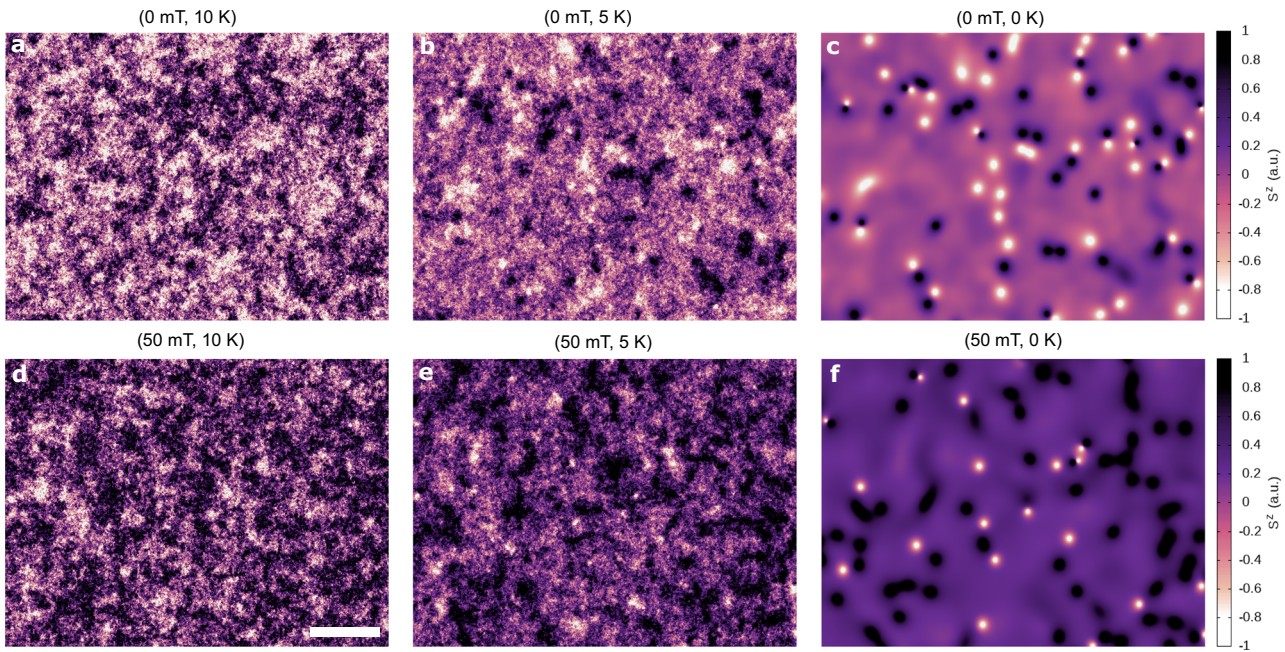

**Fig. 1 Nucleation of merons and antimerons during the cooling process. a–c** Dynamical spin configurations obtained at different temperatures ($T$) showing the evolution of the domain structure and the formation of merons and antimerons during field cooling in an external field of 0.0 mT. The $S^z$ component is used to follow the evolution of the different spin textures across the crystal surface (color map). Strong spin fluctuations are observed at temperatures below the critical temperature ($T_c = 19.07$ K, see Supplementary Fig S1), which incidentally vanished as the system cools down. Localized small areas within 0 K $\leq T \leq$ 5 K correspond to spins pointed perpendicular to the easy-plane of CrCl$_3$ in different spin polarizations (e.g., +1 or −1). At 0 K, most of the magnitudes of $S_z$ are zero throughout the crystal except at well defined small spots with either $S^z = +1$ or $S^z = −1$ in their cores. The formation of merons and antimerons occur simultaneously during the time evolution without a clear preference over the nucleation site. That is, boundaries, defects or edges are not considered. **d–f** Similar to **a–c** but at an external field of 50 mT. The applied field polarizes the spin configurations resulting in less fluctuations along of $S^z$ even though with alike domain dynamics. At $T \leq$ 5 K, the merons and antimerons are still formed but with a more preferential spin polarization, e.g., darker spots. If larger magnetic fields beyond 50 mT are applied (e.g., 100 mT), a full polarization of the topological spin textures is observed with a totality of just one kind of spin polarization. For fields above 150 mT, there is no additional nucleation of merons and antimerons throughout the crystal as the spin textures outside the vortex core follows the field direction. See Supplementary Figs. S2, S3 and Supplementary Movies S1–S3 for details. Scale bar is 50 nm.

unveil their true nature, we can calculate the topological number $N$ given by[19–22]:

$$N = \frac{1}{4\pi} \int \hat{s} \cdot \left( \frac{\partial \hat{s}}{\partial x} \times \frac{\partial \hat{s}}{\partial y} \right) dx\, dy, \qquad (4)$$

where $\hat{s} = \frac{s}{|s|}$ is the three-component spin field. Equation (4) indicates a product between the vorticity of the spin textures, which is determined by the direction of the in-plane components of the magnetic moment, e.g., $\left( \frac{\partial \hat{s}}{\partial x} \times \frac{\partial \hat{s}}{\partial y} \right)$, and the out-of-plane component of $\hat{s}$. The latter is observed at the core of the vortex while the former near the perimeter aligning with the easy-plane of the material producing magnetic helicity and polarity. The obtained spin textures in the spin dynamics follow this description forming two types of distributions of the magnetization in terms of vortex (Fig. 2c, f) and antivortex (Fig. 2d, e). To characterize which kind of spin quasiparticle is present in CrCl$_3$ (e.g., skyrmions or merons), we evaluated the integral in Eq. (4) numerically for each topological spin textures to quantify $N$ (Supplementary Section 4). This integral is performed under an area sufficiently large to include the totality of the spin textures, which was converged to 20 nm × 20 nm. This ensures that no spin contributions are left outside of Eq. (4). All three components of $\hat{s}$ are considered in the integration. We found that $N = \pm 1/2$, which is indicative of merons ($N = −1/2$) and antimerons ($N = +1/2$). Such fractional values of $N$ implicate that the total charge $Q = eN$ (where $e$ is the electron charge) of the merons and antimerons is also fractional $Q = \pm e/2$[22] with $N = \pm 1/2$ being a topological

invariant[22] since a meron topologically has half the spin winding of a skyrmion. Indeed, the spin polarization of the core can also be used to identify core-up or core-down vortex and antivortex, which differs from skyrmions and antiskyrmions[18]. This gives the possibility to identify additional degrees of freedom for magnetic helicity and polarity[18].

One of the main implications for spin textures to have a well defined topological number is that they should keep their magnetic ordering regardless of external perturbations[18] or even form a lattice of stable vortex–antivortex spin textures[13,20]. Nevertheless, by observing the dynamics of the vortex and antivortex in CrCl$_3$ at longer times (Supplementary Movies S1 and S2), we notice that both spin quasiparticles interact and disappeared through collisions, which happened roughly 15–20 ns after they are created below 5 K. This lifetime is at least two orders of magnitude larger than that previously observed for merons and antimerons in thin ferromagnetic iron layer[23], kagome magnet[24] and ferromagnetic permalloy disks[25], which suggested more stability for measurements.

Figure 3 shows that the collision process is independent of the core polarization of the meron or antimeron considered (Fig. 2d–g) but it involves at least one vortex and one antivortex during the process. That is, collisions between pairs of the same type, such as vortex–vortex or antivortex-antivortex are not observed at any temperature and magnetic field, which is in agreement with magnetic vortice theories[26]. Intriguingly, the vortex and antivortex with different core polarizations (Fig. 3a–c) approach each other in spiral orbits rotating the in-plane

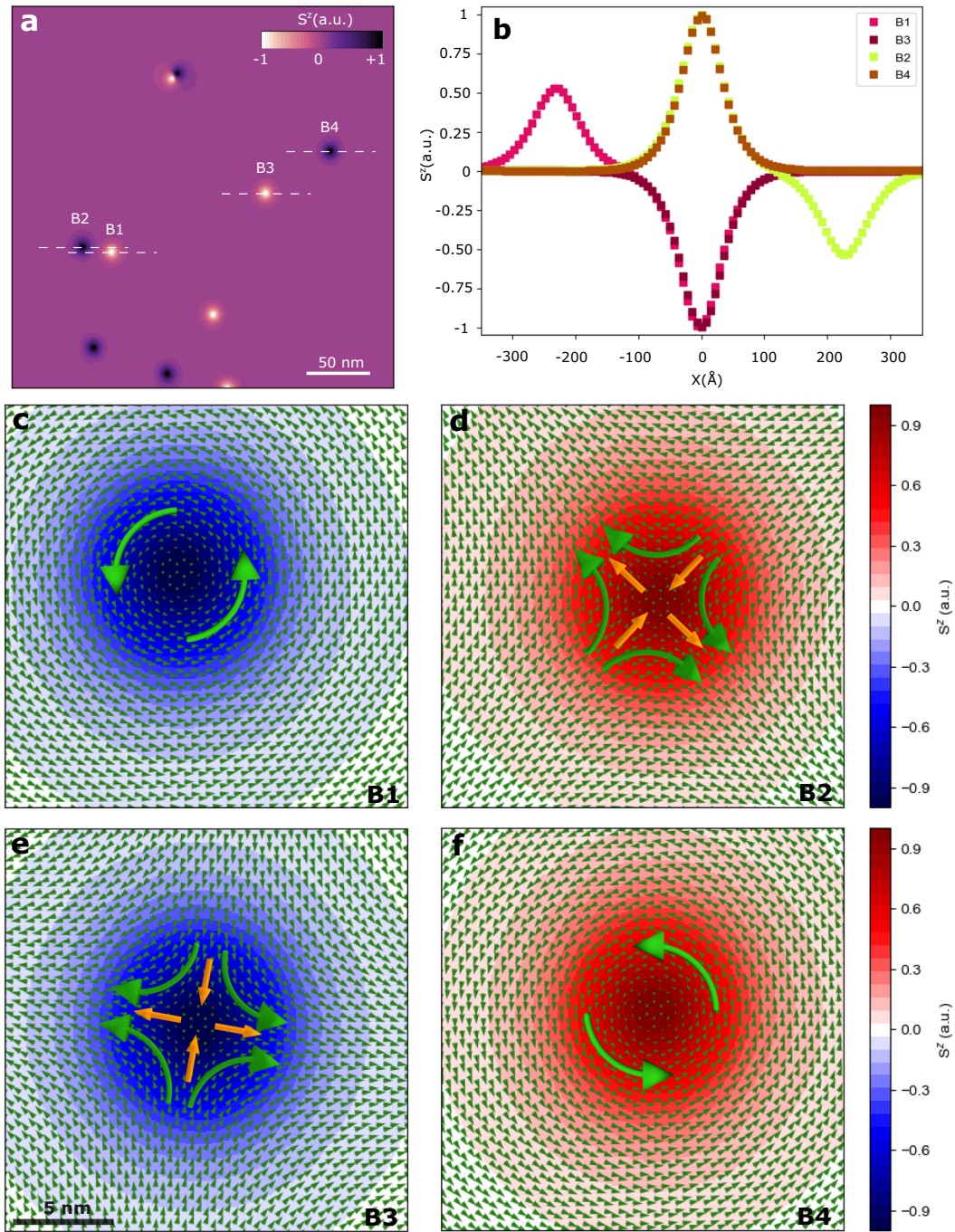

**Fig. 2 Characterizing the spin features of merons and antimerons. a** Snapshot of a spin configuration projected along the $S^z$ component (color map) at a selected time with the formation of merons and antimerons at zero magnetic field and 0 K. Some of the topological spin textures are marked as B1–B4 to highlight their features. **b** Profile of $S^z$ along the dashed lines in **a** over the distinct spin quasiparticles (B1, B2, B3, B4) showing their widths in Å's. B1 and B2 are close enough to feel the opposite spin polarization from each other. The largest peak at 0 Å is centered at the center of the meron or antimeron. B2 and B4 have both positive spin polarization with similar peak magnitude at 0 Å ($S^z = +1$), which is the opposite of B1 and B3. **c–f** Spin textures of the different quasiparticles stabilized in monolayer $CrCl_3$. Merons (**c**, **d**) and antimerons (**e**, **f**) can be determined by the topological number $N$ (Eq. (4)), which involves the vorticity (±1) and the core polarization. B1 and B4 have the same vorticity of +1 even though they are meron ($N = -1/2$) and antimeron ($N = +1/2$), respectively. Similar argument applies to B2 and B3, that is, both have vorticity of −1 but are meron and antimeron, respectively. Small arrows (green) indicate the direction and magnitude of the in-plane magnetization relative to the core (zero value). The underneath color gradient shows the variation of $S^z$ around the spin textures. It reaches its maximum magnitude at the core of the merons and antimerons. Large arrows (orange and green) give the average behavior observed around the vortex core by the in-plane magnetization.

magnetization to out-of-plane. This stabilizes new cores for both spin textures following the original core polarization at large separations. At close distances (roughly 2 lattice constants) both vortex and antivortex reduced gradually dissolving the complicated spin arrangement into the easy-plane of $CrCl_3$ with an emission of a spin-wave[27,28] (Supplementary Movie S4). In the case of merons and antimerons with similar core polarization (Fig. 3d–f and Supplementary Movie S5), there is an enlargement

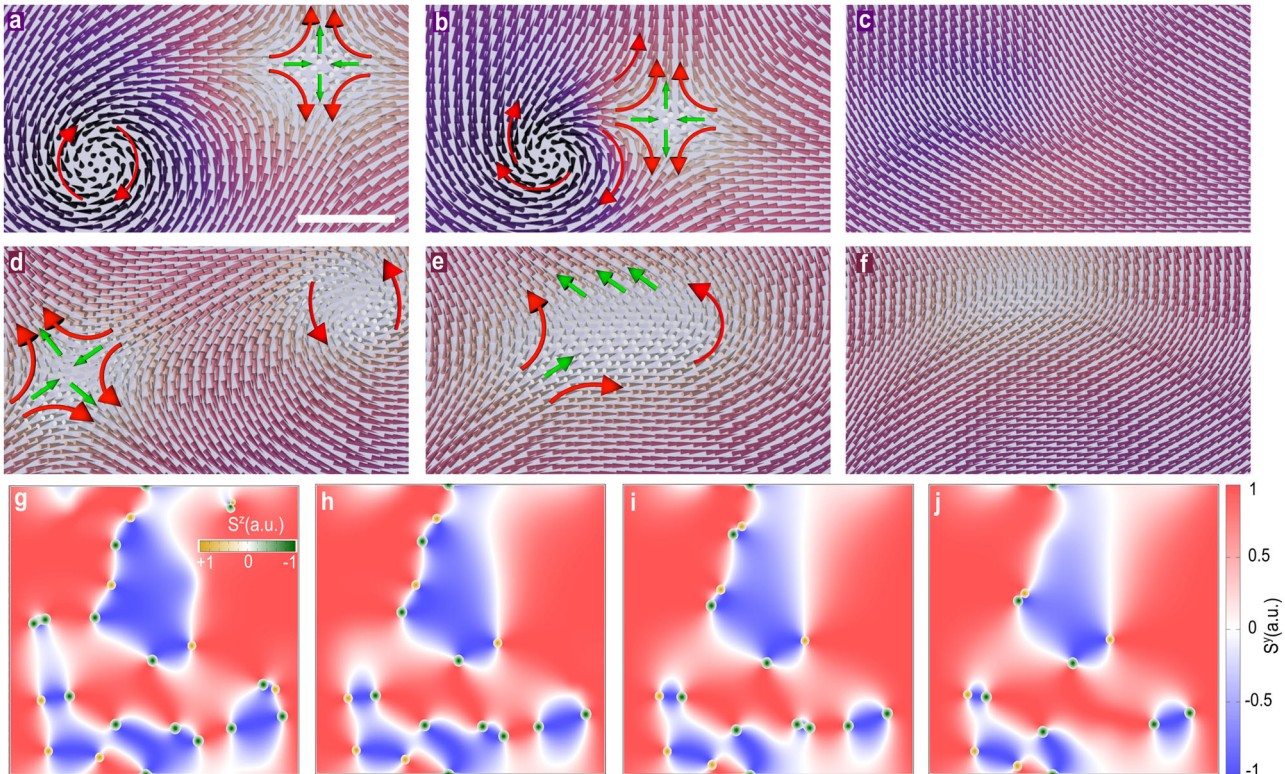

**Fig. 3 Meron and antimeron collision. a–c, d–f** Snapshots of the vortex and antivortex dynamics with antiparallel and parallel core polarization, respectively, before (**a**, **d**), during (**b**, **e**), and after (**c**, **f**) the collisions at zero field and 0 K. The dark and bright backgrounds indicate a more out-of-plane magnetization at the core of the vortex and antivortex. The big arrows in **a**, **b**, **d**, and **e** indicate the average behavior of the in-plane spin components in the perimeter. In all considered topological spin textures, similar collision scenarios are observed, which take roughly between 0.57 ns (**a–c**) and 0.17 ns (**d–f**) to occur. Scale bar is 4 nm. **g–j** Macroscale magnetic domains (blue and red) at different times showing the evolution of the merons and antimerons (small circles in green and orange) at the boundary between magnetic domains. The $S^y$ component (color map) of the magnetization is utilized to show the magnetic domain structure whereas $S^z$ for the vortex and antivortex textures. The white boundary near where the merons and antimerons are localized have $S^y = 0$ and $S^x = 0$ (Supplementary Fig. S6). $S^z$ reaches its maximum magnitude at the center of the spin textures (inset color map in **g**). The dynamics of the domains is directly coupled to the motion of the merons and antimerons, and vice-versa. At sufficient longer times, the entire system results in a mono-domain throughout the surface.

of the area with out-of-plane magnetization as both topological quasiparticles approach each other. This induces the formation of a composed spin texture with a larger extension over the surface (Fig. 3e), which incidentally dissolves to an in-plane orientation at later times (Fig. 3f). Both dynamics involving merons and anti-merons of parallel or antiparallel core polarization are intrinsically coupled to the domain structure present in CrCl$_3$ being formed shortly (~0.1−0.2 ns) after the formation of the magnetic domains. Figure 3g–j and Supplementary Movie S6 show a broad perspective of the magnetic domains as the collisions happen involving different spin textures (small dots). We noticed that both merons and antimerons are localized at the boundary between magnetic domains with different spin polarization (e.g., $S^z = +1$ and $S^z = -1$) where their displacement is along the domain edges as $S^y$ is zero. We have not observed any factor or physical ingredient that would help to determine the distance between the merons and antimerons as they appear spontaneously in the zero-field cooling. The distance between the spin textures in this sense follows the spin dynamics of the system being stochastic as our calculations indicated. Similar configurations are also observed at another in-plane component of the magnetization, e.g., $S^x = 0$, where the topological spin textures are localized (Supplementary Fig. S6). We can understand this type of magnetic frustration in terms of the competing exchange interactions between the different nearest neighbors (1st, 2nd, 3rd) in CrCl$_3$[16]. In terms of the isotropic exchange ($J_{1st}$, $J_{2nd}$, $J_{3rd}$), we observe that

the frustration takes places due to the third-nearest-neighbor, which has an anti-ferromagnetic exchange ($J_{3rd} = -0.025$ meV) relative to the first- ($J_{1st} = 1.28$ meV) and the second-nearest neighbors ($J_{2nd} = 0.072$ meV). In terms of the anisotropic exchange ($\lambda_{1st}$, $\lambda_{2nd}$, $\lambda_{3rd}$), the competition between second- and third-nearest neighbors ($\lambda_{2nd} = -0.0097$ meV, $\lambda_{3rd} = -0.0051$ meV) with the first-nearest neighbors ($\lambda_{1st} = 0.020$ meV) induced that the in-plane spins ($S^x$, $S^y$) become negligible. In addition, a full in-plane spin polarization along $S^x$ and $S^y$ would lead to a singularity of the exchange energy, which is avoided as $S^z$ becomes non-negligible[28]. This indicates that the intersections where both in-plane components of the magnetization converge to zero are an efficient environment to localize non-trivial topological spin textures. Such localization of magnetic vortices are typically seen in singly connected samples with the magnetic flux occurring at the junctions of magnetic domains. For instance, in permanent magnets[29–31] and in soft-magnetic nanodisks[32] with strong cross-tie domain-wall structures and geometry play a critical role in the generation of the vortices. Nevertheless, we find that this is not the case for monolayer CrCl$_3$. There are no structural constraints in the stabilization of the magnetic domains and the low magnetic anisotropy would orientate the spins more freely without a pre-ferential orientation within the easy-plane[33]. Therefore, such spontaneous formation of merons and antimerons is an aston-ishing, previously unreported phenomena in the magnetism of any 2D vdW magnet.

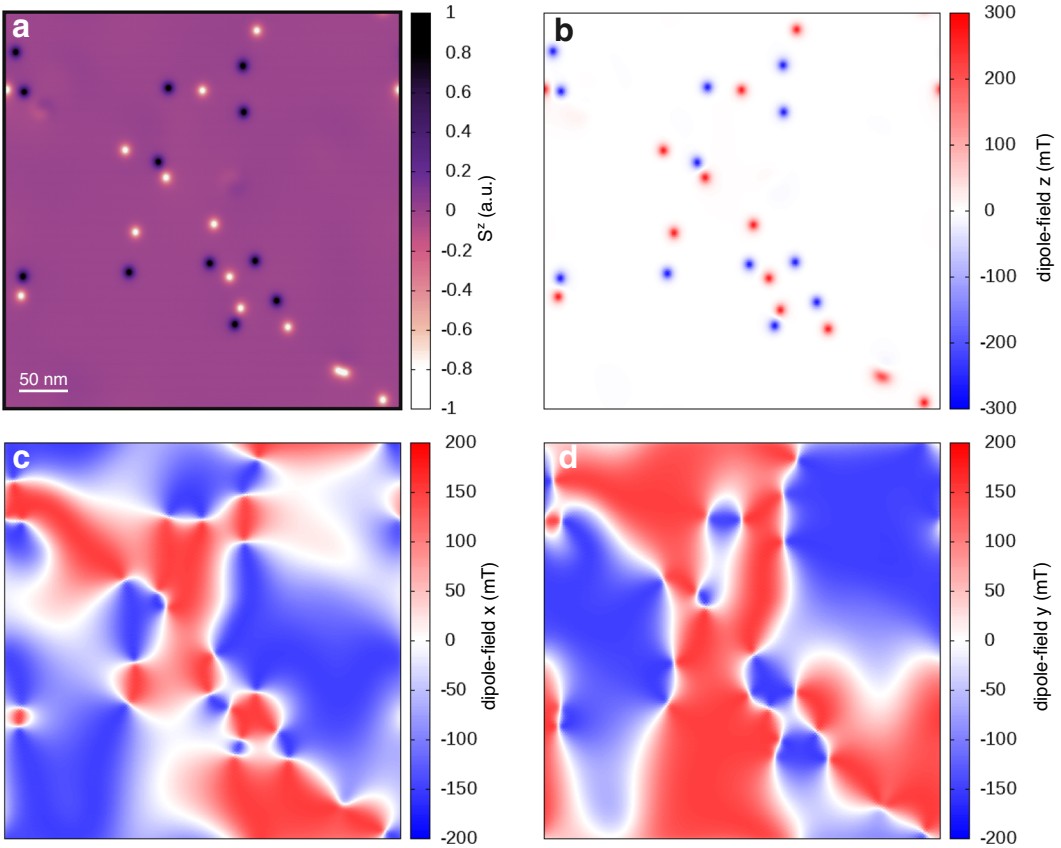

**Fig. 4 Dipolar interactions driven the formation of merons and antimerons. a** Snapshot of one of the spin dynamics at 0 K and zero magnetic field showing the out-of-plane spin component $S^z$ (color map) throughout the surface of monolayer $CrCl_3$. **b–d** Projection of the dipole–dipole interactions along of $z$, $x$, and $y$ directions, respectively, on the snapshot in **a**. The dipole fields are quantified in mT with positive (red) and negative (blue) magnitudes in the color scale. The scale bar of 50 nm is common to all panels.

An intriguing question that raised by the presence of these topological spin textures in $CrCl_3$ is what their physical origin. It is known that strongly inhomogeneous magnetic textures can be created due to the competition between local interactions, e.g., exchange and magnetic anisotropy, and long-range interactions mediated by demagnetizing fields and magnetic dipoles[18,34–37]. In the case of $CrCl_3$, the interplay between dipolar interactions and magnetic anisotropy is one of the main ingredients in the creation of merons and antimerons. Figure 4a, b shows that as the cooling process takes place, the different spin textures intrinsically carry a large component of the dipole moment perpendicular to the surface (±300 mT). This is mainly localized at the core of the quasiparticle and assists in stabilizing a strong component of $S^z$. Indeed, there is almost no difference between the projection of the magnetization perpendicular to the surface and that for the dipole-field along $z$ (Fig. 4a, b). This indicates a close relationship between dipole–dipole interactions and magnetism in $CrCl_3$. The distinct spin polarization of the core of the merons and antimerons does not give any significant variation on the magnitude of the dipole-field, which is larger than those within the in-plane components (Fig. 4c, d). Surprisingly, both components of the dipole-field ($x$ and $y$) reach smaller magnitudes (±200 mT) than those along $z$ throughout the surface, and are strictly zero at the position of the spin textures.

In this context, the in-plane dipole fields favor an in-plane magnetization whereas the single-ion anisotropy orients the spins perpendicular to the surface. In $CrCl_3$, the magnetic anisotropy is small, which allows most of the spins to follow the dipolar directions except those at the core of the merons and antimerons. In such particular locations, the stronger $z$ component of the

dipole-field pushes the spins out-of-plane enhancing the magnetic anisotropy. In any other part of the surface without the topological spin textures, the magnetization rotates parallel to the surface as an effect of the dipolar interactions[33]. It is worth mentioning that no transition between a previously stabilized magnetic configuration, such as stripes, into the non-trivial magnetic textures, is noticed in the spin dynamics with or without an applied field as it has been suggested as a potential origin of bubbles or skyrmions[38–40]. Moreover, we do not take into account asymmetric exchange (Dzyaloshinskii–Moriya interactions) into our simulations, which was initially checked to have no effect on the dynamics of the merons and antimerons (Supplementary Fig. S14). This excludes additional mechanisms based on relativistic effects[18]. We also considered simulations without the inclusion of dipolar fields (Supplementary Fig. S7) for $CrCl_3$ with a two-fold implication. First, there is no appearance of non-trivial topological spin textures as the magnetization is consistently out-of-plane over the entire crystal. Second, there is the stabilization of an easy-axis perpendicular to the surface following the single-ion anisotropy. That is, there is a suppression of the easy-plane experimentally observed for $CrCl_3$[33]. Even though other models[36,37] utilized for magnetic thin films assumed that the dipole–dipole directions can be effectively replaced by an easy-plane (XY), our calculations indicate that the inclusion of dipolar interactions plays a key role in the description of the magnetic properties of $CrCl_3$ (Supplementary Fig. S11). The spins textures formed at such artificial easy-plane[36,37] looked more chaotic than those computed without such restriction being more complex to assign any clear feature or to determine a topological number $N$.

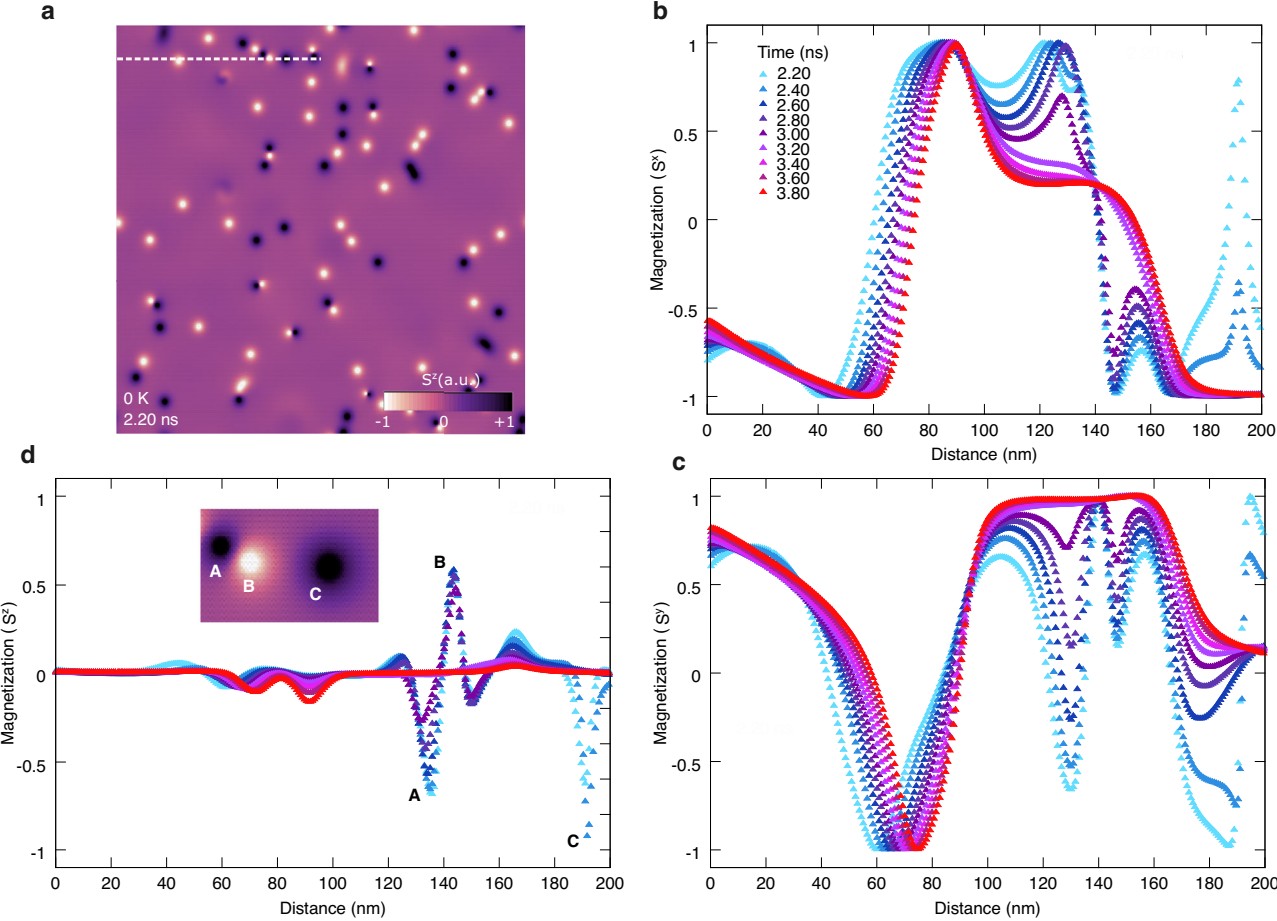

**Fig. 5 Spin fluctuations-driven magnetic domain metastability. a** Snapshot of a spin dynamics of monolayer $CrCl_3$ obtained through zero-field cooling after 2 ns and reaching 0 K. The magnetization perpendicular to the surface ($S^z$) is displayed showing the formation of merons and antimerons (small dots). Bright (dark) areas correspond to $S^z = \pm 1$, respectively, in the color scale. A path (dashed line) of 200 nm is drawn to show the spatial variation of the magnetization at different times ($t \geq 2.20$ ns). **b–d** Variations of the in-plane components of the magnetization ($S^x$, $S^y$) and $S^z$, respectively, along of the path shown in **a** within 2.20–3.80 ns after 0 K is reached. The inset in **d** shows a small area from **a** along the path with the formation of the merons and antimerons. The corresponding variation of $S^z$ at different times at A, B, and C is also showed.

We also noticed that the inclusion of biquadratic exchange and next-nearest neighbors is critical for the stabilization of non-trivial topological spin textures in 2D magnets (Supplementary Figs. S12 and S13). For the former, the higher-order exchange between the spins gives further stability to Eq. (1) since the sign of $K_{ij} = 0.22$ meV is positive[16]. For the latter, the non-inclusion of second- and third-nearest neighbors in Eq. (1) resulted in the absence of vortex or antivortex spin textures even if dipolar interactions are considered. This can be reasoning in terms of the large contributions of the in-plane anisotropic exchange from the second- and third-nearest neighbors ($\lambda_{2nd} + \lambda_{3rd} = -10.25$ μeV) relative to the first-nearest neighbors ($\lambda_{1st} = 20.07$ μeV). This indicates that although the dipolar interactions can assist in the creation of an easy-plane over the surface, they are not sufficient to polarize an in-plane magnetization ($S^x$, $S^y$), which is greatly affected by the next-nearest-neighbor exchange interactions. Furthermore, we have extended our simulations for other materials in the Cr-trihalide family $CrX_3$, X=F, Br, I (Supplementary Section 5 and Supplementary Movies S7–S9). This will allow us to have a broader perspective of whether other layers in the same family, and in a similar chemical environment (e.g., halogens) and symmetry (e.g., honeycomb), would be susceptible to the creation of merons and antimerons. We found no evidence of non-trivial spin textures as those in $CrCl_3$. There is a continuous evolution of the magnetic domains from high

temperatures till 0 K with no formation of complex spin configurations. We can understand these results in terms of the strong out-of-plane single anisotropy[16] present in $CrX_3$ (X=F, Br, I) with the dipolar interactions also pointing perpendicular to the surface. This indicates that the presence of an easy-plane is an important requisite for the stabilization of non-trivial spin quasiparticles.

The complex dynamics of the merons and antimerons in $CrCl_3$ can be directly related to the magnetic stability of the layer[26]. Strong thermally driven spin fluctuations prevent a clear observation of the distinct spin textures, which are well pronounced for $T \leq 5$ K (Fig. 1). Such spin fluctuations are due in part to the low single-ion anisotropy ($D \sim 0.01$ meV) of $CrCl_3$, which would require low-energy excitations to change the orientation of the surface spins. Other contribution factor is the metastability of the magnetic domains in $CrCl_3$ that continuously evolve as a function of time. For magnetic compounds with strong easy-plane anisotropy, demagnetization or long-range dipolar fields are known to affect the ground-state of topological spin textures[41,42], which an equilibrium state is normally achieved beyond the field cooling process ending at 0 K. Although any thermal contribution to the magnetic domains will be zero at this limit, the spins would still evolve to stabilize the ground-state via the minimization of other contributions of the total energy, e.g., exchange, anisotropy. This process can be observed in Fig. 5a, for the time evolution of one of

the spin dynamics of monolayer $CrCl_3$ once the system had achieved 0 K within 2.0 ns at zero field. There is a continuous modification of the domain-wall profiles through all components of the magnetization ($S^x$, $S^y$, $S^z$) over time. The variations on $S^x$ and $S^y$ across the magnetic domains (Fig. 5b, c) tend to be broader with less peaky changes as those observed along $S^z$ due to the presence of merons and antimerons (Fig. 5d). For them, several sharp changes appeared and vanished on a time scale of few tenths of nanoseconds indicating the stochastic nature of the spin fluctuations in the system. Indeed, we observed such random fluctuations of the magnetization even beyond 4 ns, which suggests that the system may not be in a local minimum but rather at a flat energy landscape. As a matter of fact, an increment of $D$ will not cease the fluctuations as our calculations showed that they may be intrinsic to $CrCl_3$ (Supplementary Fig. S15).

## Discussion

The discovery of non-trivial topological spin textures (merons and antimerons) in a non-chiral 2D magnetic material ($CrCl_3$) opens the possibility for other layered materials display such behavior. Some guidelines for looking into materials that may develop such quasiparticles would be (i) a weak out-of-plane single-ion anisotropy ($D_i$), (ii) high in-plane dipolar interactions, and (iii) competition between next-nearest neighbors. Our simulations indicate that the general nature of the formation of merons and antimerons in a 2D magnet is due to the combination of these three factors as cooling occurs. The delicate balance between anisotropy, dipole–dipole interactions, and exchange competition has a major effect in the stabilization of the core vortex, the perimeter, and the spin helicity as well as the polarity of the non-collinear spin textures. Our simulation results also indicate the observation of meron and antimeron spin textures at no applied magnetic field, low temperatures and without edge effects[18,43]. Thus, the problem now turns in the search of other layered compounds where such guidelines could be fulfilled. Efforts on the discovery of novel magnetic sheets that hold topological non-trivial quasiparticles will pass through the accurate calculation of magnetic parameters, and subsequent atomistic simulation of large-scale properties (i.e., magnetic domains, domain walls). Furthermore, interactions with a substrate are also important to be considered. Even though we have not considered them explicitly in our calculations, different support could either enhance or deteriorate the magnetic ordering of thin layered materials. Progress in the isolation of $CrCl_3$ from spurious support interactions have recently been reported[44]. Such experiments where large-scale monolayer $CrCl_3$ are grown via molecular beam epitaxy on graphene/6H-SiC(0001) substrates provided a new avenue for the validation of the predictions included in this study. In addition, merons and antimerons are likely to be created under ultrafast laser excitations[25] and current pulses[45], which open the door for validating our predictions. With the prompt integration of magnetic layered materials in device platforms and the emergence of more compounds with similar characteristics, it is a matter of time till experimental realization and subsequent control will be achieved for such non-trivial spin topology. In this sense, the emergent electrodynamics initially established for skyrmions[46] can be explored further at a more fundamental level using merons in a truly 2D magnet.

## Methods

All methods are included in Supplementary Information, which includes Supplementary Sections S1–S5, Supplementary Movies S1–S9, and Supplementary Figs. S1–S14.

## Data availability

The data that support the findings of this study are available within the paper and its Supplementary Information.

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

## Acknowledgements

E.J.G.S. thanks Dina-Abdul Wahab for assistance in the preparation of Fig. 5 and Supplementary Figs. S8–S10. R.F.L.E. gratefully acknowledges the financial support of the Engineering and Physical Sciences Research Council (Grant No. EPSRC EP/P022006/1) and the use of the VIKING Cluster, which is a high performance compute facility provided by the University of York. This work was enabled by code enhancements to the VAMPIRE software implemented under the embedded CSE program (ecse0709) and (ecse1307) of the ARCHER UK National Supercomputing Service. E.J.G.S. also acknowledges computational resources through the UK Materials and Molecular Modeling Hub for access to THOMAS supercluster, which is partially funded by EPSRC (EP/P020194/1); CIRRUS Tier-2 HPC Service (ec131 Cirrus Project) at EPCC funded by the University of Edinburgh and EPSRC (EP/P020267/1); ARCHER UK National Supercomputing Service (http://www.archer.ac.uk) via Project d429. EJGS acknowledges the EPSRC Early Career Fellowship (EP/T021578/1) and the University of Edinburgh for funding support.

## Author contributions

E.J.G.S. conceived the idea and supervised the project. M.A. performed ab initio and Monte Carlo simulations under the supervision of E.J.G.S. S.J. implemented the dipole approximations. M.A. and E.J.G.S. elaborated the analysis with inputs from R.F.L.E. and K.S.N. E.J.G.S. wrote the paper with inputs from all authors. All authors contributed to this work, read the manuscript, discussed the results, and agreed to the contents of the manuscript.

## Competing interests

The authors declare no competing interests.
