## [Peer Review File · Nature Communications]

Reviewers' Comments:

Reviewer #1:

Remarks to the Author:

The authors have investigated an atomistic spin model of monolayer CrCl₃ and find that, upon cooling from above a Curie temperature, meron and anti-meron structure emerge. The atomistic Heisenberg model uses parameters obtained from DFT calculations and includes a smoothed long-range dipolar field.

In general the work is carefully carried out and well described, although the English is slightly awkward at times. My main comment about the manuscript is the following: It does not seem to me that the results are in any way specific to CrCl₃ - I can imagine lots of continuum 2D magnetic systems with some weak out-of-plane anisotropy and some in-plane frustration, and I imagine I would get similar results. So what in this work is specific to CrCl₃? The authors removed the dipolar field and showed that would drive the magnetization out of plane. That makes sense intuitively but is again kind of generic to 2D continuum magnets. The authors have included up to third-nearest neighbor interactions and biquadratic interactions. Are all these needed? Aren't the important ingredients long-range dipolar fields (demagnetizing fields) and some frustration? Incidentally, the authors have not at all described the geometry of the CrCl₃ system - I would imagine that the frustration arises from the triangular nets but the authors only once mentions frustration and does not in any way the geometry to frustration and creation of merons/antimerons.

Also, I am not so sure about the language the authors use, such as "One of the main implications for the spin texture to have a well defined topological number is that they should be topologically protected against external perturbations" (p. 7). What does that mean? Strictly speaking, integer Pontryagin indices are topological invariants of a 3D continuous vector field on a 2D manifold. I do not think a topological charge of 1/2 is strictly speaking a topological invariant - but 0 or +/- are. What muddles things is that here we are dealing with a vector field on a lattice, and "protection" comes from an energy scale, typically the exchange energy cost if one tries to destroy a meron or vortex.

In the introduction, the authors also talk about pair creation of meron-antimeron pairs in a random lattice. I don't understand that statement. What is the "random lattice"? Isn't the underlying model a Heisenberg model on a regular lattice?

In summary, the work is potentially interesting but it is in its present form leaving many important questions unanswered. In my opinion, the manuscript needs a major revision before I would consider recommending it for publication in Nature Comm.

Reviewer #2:

Remarks to the Author:

I have read and carefully analyzed the manuscript entitled "Lifetime evolution of meron and antimeron topological spin textures in the two-dimensional magnet CrCl₃" by Augustin et al.

The authors claim that their results are of great interest because of the recently discovered magnetism in the vdW material CrCl₃ [reference missing in abstract], there are no comparisons to comparisons to experimental findings. This is purely theoretical work in which material parameters (that have been published elsewhere) are used to simulate the magnetization in a 2d setup at finite temperatures in state-of-the-art model and with methods are well-known and established. The movies are beautiful to look at but the analysis is disappointing (see below for a few examples). Finally, the authors summarize that merons and antimerons might be created stochastically when cooling down from large temperatures but that they will decay via pair

annihilation. This conclusion seems to me somewhat trivial as of course defects of all kind might be stabilized when quenching a system and the pair creation and annihilation of merons (or vortices, skyrmions, etc) is also well-known (including a Nobel prize?). The lifetime, mentioned in the title, is not discussed in this manuscript.

I therefore do not see any scientific advance in this report and have concerns about the interpretation of some findings. I can not recommend it for publication -- in particular not in a journal with a high impact as Nature Communications.

In particular concerning the analysis of the simulations, I have a number of questions and serious concerns.

(1) The topological charge in Eq.4 is only quantized to integers if the integration area is a closed surface. Alternatively, the boundary of the area can be "trivial" in some sense, e.g., polarized. None is the case here, therefore the charge should depend on the integration area (which the authors do not specify). Moreover, the charge cannot be quantized to $\pm 1/2$ but only to integers. So this analysis is very questionable. Also the explanation of the formula itself is very questionable, e.g., \hat{s} is of course the magnetic moment and not only its out-of-plane component, as claimed by the authors.

(2) The authors continue in line 115 that topology should imply stability which is not true, in particular when the antiparticle is present in the system.

(3) The authors notice the spiraling orbit during collision, lines 123 and following, which is also well-known for vortices and can be described simple analytical arguments that use collective coordinates. In fact, a lot of findings that are reported in this paper are not new but references and a maybe deeper analysis are missing. For example, the long-range interaction between defects in this setting would be interesting to draw conclusions on the lifetime and also the role of the biharmonic interaction would be interesting to know.

(4) The authors compare their simulations with dipolar fields to simulations without dipolar fields but otherwise the same parameters. That obviously does not yield inplane magnetizations as the very strong inplane anisotropy by the dipolar interaction is missing. In studies of thin magnetic layers or monolayers, this is usually corrected by approximating the dipolar interaction in monolayers by an effective local inplane anisotropy which would stabilize inplane magnetic textures here. Therefore, this comparison is unvalid and should be improved. I suspect that the dipolar interactions can be entirely replaced by a local easy plane anisotropy, otherwise distortions of the antivortex should be visible because of its nonzero magnetic charges (which are also not analyzed in this manuscript).

(5) Concerning the presentation of the results, I don't understand why all spin components use the same color code (which is confusing) while sometimes different color codes are used for the same quantity (see Fig.S3). Do the authors consider variations of the spin length (this is not explicitly mentioned)? If not, a color scheme that covers the 2-sphere could be much more easy to understand for the reader. I also don't understand how the simulation cell looks like. What is the lattice underlying all those figures? This is not clear to me from the text in the supplement.

Reviewer #3:

Remarks to the Author:

The manuscript by Augustin et al. describes the formation and annihilation of meron-type topological spin structure in CrCl₃, a currently fashionable two-dimensional magnet. The work is based on an extensive simulation formalism involving Monte Carlo and LLG spin dynamics calculations. The predictions are made for a monolayer sheet of CrCl₃.

The results presented are very interesting and shed some light on the micromagnetism in v.d.W. materials. The CrCl₃ sheet is shown to host merons and antimerons, which spontaneously form below the theoretically calculated Curie temperature (around 19 K). The dynamic behaviour of the

topological structures is analysed as a function of temperature and magnetic field, showing that they eventually annihilate resulting in a homogeneous magnetisation. In my view the paper is concise, but still leaves some open questions, which should be answered before publication can be considered. The questions are listed below.

1. The spin Hamiltonian which forms the basis for the calculations does not contain - apart from a small single ion anisotropy - any further anisotropies or terms related to spin-orbit coupling. For a free-standing CrCl₃ monolayer this approach may be justified. In any realistic experiment the CrCl₃ sheet will have to be supported by a substrate, which may introduce additional interactions. To what extent will an additional exchange or spin-orbit interaction with the substrate affect the results?
2. The movies mapping the cooling down process seem to indicate that even at temperatures far below T_c (temperature window 5 - 20K) thermal fluctuations suppress the meron/antimeron formation. What is the reason for this strong influence of fluctuations? One reason mentioned is the small single-ion anisotropy. Would an increase of the SIA stabilise the situation? Could this be a prediction for other material systems beyond CrCl₃ to show similar phenomena?
3. The nucleation of merons and antimerons appears to happen only on domain boundaries. The distance between neighbouring topological structures varies a lot from event to event. Why are the domain walls preferred and what determines the distance between the merons?
4. Do the domain patterns and topological features form at the same time during the cool down process or is there a time lag between them? This is difficult to tell from the movies.
5. In their discussion of the vortex-antivortex annihilation process (p.7) which results in the emission of spin waves the authors cite Ref. 28. However, this reference deals with the switching of a vortex core. Hertel et al. (PRL 2006) showed that this process involves a complex intermediate state with two additional vortex cores of opposite polarity created, one of which annihilates with the original vortex core releasing spin waves. Why do such intermediate states not seem to play a role in the meron/antimeron annihilation processes?
6. The title of the paper reads "lifetime evolution", but I don't find a quantitative determination of the lifetime for the various parameter sets studied. The movies cover a range of 40 ns. However, it is difficult to discern, for example, whether or not the strength of the applied field affects the lifetime of the merons/antimerons or just the density during the formation. Information of the actual lifetimes as a function of temperature and magnetic field would be a critical information for experimentalists. Their primary question may be: what needs to be done to bring the lifetime of the topological structures into a regime compatible with an experimental approach to measure/image them?

Response to Reviewers Comments:

Reviewer 1:

The authors have investigated an atomistic spin model of monolayer CrCl₃ and find that, upon cooling from above a Curie temperature, meron and anti-meron structure emerge. The atomistic Heisenberg model uses parameters obtained from DFT calculations and includes a smoothed long-range dipolar field.

In general the work is carefully carried out and well described, although the English is slightly awkward at times.

Response 1: We thank the Reviewer for reading our manuscript and for his/her thoughtful comments and suggestions. We have substantially expanded the results presented in our work to address all comments raised by the Reviewer and changed the manuscript accordingly. We also reviewed the English following the Reviewer suggestion. A point-by-point revision is shown below, with the location referring to the revised manuscript.

My main comment about the manuscript is the following: It does not seem to me that the results are in any way specific to CrCl₃ - I can imagine lots of continuum 2D magnetic systems with some weak out-of-plane anisotropy and some in-plane frustration, and I imagine I would get similar results. So what in this work is specific to CrCl₃?

Figure R1: Zero-field cooling simulations for monolayer CrBr₃ from temperatures above the Curie temperature (60 K) till 0 K. We use similar setup as those for monolayer CrCl₃ included in our manuscript. The out-of-plane component of the magnetization (M_z) is displayed in the plots. Dark/bright areas correspond to spins point either up or down to the surface. The scale bar is 100 nm. The magnetic parameters are from Kartsev et al. arXiv:2006.04891.

Response 2: In principle CrCl₃ reunites a set of ingredients in terms of i) exchange interactions (e.g. 1st, 2nd, 3rd nearest neighbours), ii) low out-of-plane single-ion anisotropy and iii) a high in-plane dipole-dipole interactions that make it unique for the stabilization of non-trivial spin textures as our simulations suggested. As it is commented over the next **Responses 2-5**, we provided further scientific evidences of this statement. We, however, agreed with the Reviewer that other 2D magnetic materials could develop such spin-quasiparticles following the guidelines in our manuscript. Hence, the main problem turns in finding other materials with similar characteristics. Obviously, the phase space for such exploration in terms of material parameters (e.g. exchange integrals, magnetic anisotropies, dipolar fields, etc.) is too large to be undertaken systematically and completely beyond the scope of our manuscript. Nevertheless, we can still provide additional data to hold our argument. To give a concrete example, we have extended our simulations for the other layered compounds of the CrX₃ (X=F, Br, I) family using similar approaches as those in our manuscript. This will allow us to have a broader perspective whether other materials in a similar chemical environment (e.g. halogens) and symmetry (e.g. honeycomb) would be

susceptible for the creation of merons and anti-merons. **Figures R1-R3** and **Movies R1-R3** show that as the cooling down takes place in CrF_3 , CrBr_3 and CrI_3 none of these compounds show any sign of non-trivial topological spin textures (e.g. skyrmions, merons). There is a continuous evolution of the magnetic domains from high temperatures till 0 K with no formation of complex spin configurations.

Modifications: We have included these new results (**Figures R1-R3**, **Movies R1-R3**) in the manuscript with additional discussions in the main text (pages 12-13, lines 225-235), Supplementary Figures S8-S10, and Supplementary Movies S7-S9.

Figure R2: Zero-field cooling simulations for monolayer CrI_3 from temperatures above the Curie temperature (64 K) till 0 K. We use similar setup as those for monolayer CrCl_3 included in our manuscript. The out-of-plane component of the magnetization (M_z) is displayed in the plots. Dark/bright areas correspond to spins point either up or down to the surface. The magnetic parameters are from Kartsev et al. arXiv:2006.04891.

Figure R3: Zero-field cooling simulations for monolayer CrF_3 from temperatures above the Curie temperature (XX K) till 0 K. We use similar setup as those for monolayer CrCl_3 included in our manuscript. The out-of-plane component of the magnetization (M_z) is displayed in the plots. Dark/bright areas correspond to spins point either up or down to the surface. The magnetic parameters are from Kartsev et al. arXiv:2006.04891.

The authors removed the dipolar field and showed that would drive the magnetization out of plane. That makes sense intuitively but is again kind of generic to 2D continuum magnets.

Response 3: As discussed in **Response 2**, our simulations suggested a general mechanism for the formation of non-trivial spin textures in a 2D magnetic material. Nevertheless, we can not state without the support of more calculations whether other layers could develop similar behaviour. The fine interplay between various ingredients such as exchange, anisotropy and dipole-dipole interactions, play a major role in the creation of merons. See **Response 4** below. Therefore, the problem now becomes to find other 2D magnetic materials with similar magnetic environment which might allow the creation of non-trivial topological spin-textures. In this aspect, our manuscript opens novel avenues for investigations in this issue.

Modifications: We have included a new section in the manuscript titled "Implications and Prospects" (pages 14-15, lines 259-280) where we provided further comments and analysis.

The authors have included up to third-nearest neighbor interactions and biquadratic interactions. Are all these needed? Aren't the important ingredients long-range dipolar fields (demagnetizing fields) and some frustration?

Response 4: The spin Hamiltonian showed in Eq. 4 is a culmination of a careful and deep analysis among several possible models to describe the magnetic properties of 2D vdW magnets (see Ref. 23 for additional details). We found out at early stages of this research that the inclusion of second- and third-nearest neighbours is critical for the stabilization of topologically non-trivial spin textures. **Figure R4** shows that the inclusion of only first-nearest neighbours on bilinear (J_{1st} , λ_{1st}) and biquadratic (K_{1st}) exchange interactions, and dipolar interactions is not sufficient for the description of the magnetic structure of CrCl_3 . That is, we do not observe the formation of any vortex and/or antivortex over the evolution of the spin dynamics.

Conversely, we also checked whether K_{1st} is important for the stabilization of meron/antimerons spin-textures, but still considering dipolar interactions and up to third-nearest neighbours on bilinear exchange (J_{1st} , J_{2nd} , J_{3rd} , λ_{1st} , λ_{2nd} , λ_{3rd}). **Figure R5** shows that without biquadratic exchange the vortex and antivortex spin-textures become noisier with a more chaotic pattern over the surface. This indicates that the inclusion of K_{1st} into Eq. 4 gives further stability of the vortex and antivortex.

Moreover, the inclusion of up to third-nearest neighbours also assists in the creation of frustration and directly related with the dynamics of the merons and antimerons over the domain structures. **Response 5** below provided additional arguments for this point.

In summary, for the stabilization of spin-textures in CrCl_3 the interplay between exchange interactions, dipolar fields and magnetic anisotropy is the key behind their formation. Thus, we cannot assign just one main factor without considering the others.

Modifications: We have included the new results in **Figures R4-R5** in Supplementary Figures S12 and S13, and additional discussions in the main text following this response (page 12, lines 220-230).

Figure R4: Snapshot of a spin dynamics of CrCl_3 at different temperatures (10 K, 5 K, 0 K) without considering second- and third-nearest neighbours (J_{2nd} , J_{3rd} , λ_{2nd} , λ_{3rd}) but including dipolar interactions and biquadratic exchange into the simulations. There is no formation of vortex or antivortex spin textures throughout the system at this level of theory.

Figure R5: Snapshot of a spin dynamics at 5 K of monolayer CrCl_3 without considering biquadratic exchange interactions, but including up to third-nearest neighbours on bilinear exchange, and dipolar interactions.

Incidentally, the authors have not at all described the geometry of the CrCl_3 system - I would imagine that the frustration arises from the triangular nets but the authors only once mentions frustration and does not in any way the geometry to frustration and creation of merons/antimerons.

Response 5: The geometry is described in Supplementary Figure S5 for the honeycomb lattice formed by the Cr atoms in a monolayer CrCl_3 . In terms of the isotropic exchange (J_{1st} , J_{2nd} , J_{3rd}), we observe that the frustration takes places due to the third nearest-neighbour which has an anti-ferromagnetic exchange ($J_{3rd} = -0.025$ meV) relative to the first- ($J_{1st} = 1.28$ meV) and the second-nearest neighbours ($J_{2nd} = 0.072$ meV). In terms of the anisotropic exchange (λ_{1st} ,

λ_{2nd} , λ_{3rd}), the competition between the second- and third-nearest neighbours ($\lambda_{2nd} = -0.0097$ meV, $\lambda_{3rd} = -0.0051$ meV) with the first-nearest neighbours ($\lambda_{1st} = 0.020$ meV) induced that the in-plane spins at the intersection between magnetic domains along of S^x and S^y resulted in vanished magnetization.

Modifications: We have extended the discussions in the manuscript regarding the frustration and how it occurred in CrCl_3 (page 9, lines 156-167).

Also, I am not so sure about the language the authors use, such as "One of the main implications for the spin texture to have a well defined topological number is that they should be topologically protected against external perturbations" (p. 7). What does that mean?

Response 6: At this point in the text, this discussion referred to the possibility that topological spin-textures should not change by a continuous deformation of the field configuration. That is, they should not collapse or merge as it was observed initially in Figures 1-2. This sentence was to emphasize this aspect. However, it seems it is not totally clear.

Modifications: We have rewritten this sentence to give a clearer description of the topological protected features of the merons (page 7, lines 123-125).

and the total charge is

$$Q = \int d^2r \delta\rho(r) = \frac{1}{2} [m^z(\infty) - m^z(0)] \quad (103)$$

For a meron, the spin points up or down at the core center and tilts away from the \hat{z} direction as the distance from the core center increases. At asymptotically large distances from the origin, the spins point purely radially in the \hat{x} - \hat{y} plane. Thus the topological charge is $\pm 1/2$ depending on the polarity of core spin. The general result for the topological charge of the four meron flavors may be summarized by the following formula:

$$Q = \frac{1}{2} [m^z(\infty) - m^z(0)] n_v \quad (104)$$

where n_v is the vortex winding number. The formulas derived above for the meron charge do not rely for their validity on the variational *ansatz* assumed in Eq. (101). They are quite general and follow from the fact that a meron topologically has half the spin winding of a skyrmion. The meron charge of $\pm 1/2$ is a topological invariant and implies that the electrical charge is $\pm ve/2$.

The fact that merons carry fractional charge $\pm ve/2$ can be deduced from a Berry phase argument similar to the one used to find the skyrmion charge. We simply note that an electron moving at a large distance around a meron will have its spin rotated through 2π in the \hat{x} - \hat{y} plane due to the vorticity. We know that the Berry's phase for rotating a spin one-half object in such a way is $\exp(i2\pi S) = -1$. Thus the meron produces the same Berry's phase as half a flux quantum. From Eq. (69) we then obtain $\Delta Q = \pm ve/2$. [The ambiguity of the sign of the charge associated with half a flux quantum can only be resolved by examining the behavior of m^z in the meron core. It depends on whether the midgap state induced by the topological defect (discussed below) is empty or occupied.]

It is instructive to write down explicit microscopic variational wavefunctions for vortices (merons). We start with the simplest example: a meron with vorticity +1 and charge $-1/2$ that has the smallest possible core size:

$$|\Psi_{+1, -1/2}\rangle = \sum_{m=0}^M \left(\frac{1}{\sqrt{2}} c_{m1}^\dagger + \frac{1}{\sqrt{2}} c_{m+11}^\dagger \right) |0\rangle \quad (105)$$

Figure R6: Extract from Ref. 29 regarding the fractional charge ($\pm e/2$) of a meron and antimeron. This is a general definition obtained through the topology of their spin textures.

Strictly speaking, integer Pontryagin indices are topological invariants of a 3D continuous vector field on a 2D manifold. I do not think a topological charge of $1/2$ is strictly speaking a topological invariant - but 0 or +/- are. What muddles things is that here we are dealing with a vector field on a lattice, and "protection" comes from an energy scale, typically the exchange energy cost if one tries to destroy a meron or vortex.

Response 7: The Pontryagin index or the topological number N (Eq. 4) resulting in $N = \pm 1/2$ for the spin-textures observed in CrCl_3 is a topological invariant since a meron topologically has half the spin winding of a skyrmion, which is a topological invariant (see highlights in **Figure R6**).

A $N = \pm 1/2$ implies that the total charge of a meron or anti-meron is $\pm e/2$, where e is the electrical charge. This can be deduced from a Berry

phase argument as in the following (Ref. 29). An electron moving around a meron will have its spin rotated by 2π in the xy -plane due to the vorticity. However, the Berry's phase for a rotating a spin of one-half object is $\exp(i2\pi S)=-1$. Hence, the meron generates the same Berry's phase as half that of a flux quantum. Therefore, we are confident in the accuracy of our analysis and the results throughout our manuscript.

Modifications: We have included additional discussions in the main text regarding the charge and the computation of $N=\pm 1/2$ (pages 6-7, lines 109-122).

In the introduction, the authors also talk about pair creation of meron-antimeron pairs in a random lattice. I don't understand that statement. What is the "random lattice"? Isn't the underlying model a Heisenberg model on a regular lattice?

Response 8: In several systems developing non-trivial topological spin-textures (merons or skyrmions), these spin-quasiparticles appeared in a well-organized arrangement or lattice occupying specific sites throughout the material (e.g. Refs. 8, 19, 25). The random lattice mentioned in the introduction is meant to give a similar idea where the merons/antimerons might form an ordered pattern on CrCl_3 . However, as we showed in the manuscript, we noticed that the merons and anti-merons distribute randomly throughout the surface following the magnetic domains.

Modifications: We have re-written that sentence to clear state our arguments (page 3, lines 45-47).

In summary, the work is potentially interesting but it is in its present form leaving many important questions unanswered. In my opinion, the manuscript needs a major revision before I would consider recommending it for publication in Nature Comm.

Response: We thank the Reviewer for his/her kind words regarding our manuscript. We have carefully reviewed our manuscript and included additional results that support our findings. We believe that with these modifications our manuscript can move forward to being considered for publication in Nature Comm.

Reviewer 2:

I have read and carefully analyzed the manuscript entitled "Lifetime evolution of meron and antimeron topological spin textures in the two-dimensional magnet CrCl_3 " by Augustin et al.

The authors claim that their results are of great interest because of the recently discovered magnetism in the vdW material CrCl_3 [reference missing in abstract], there are no comparisons to comparisons to experimental findings. This is purely theoretical work in which material parameters (that have been published elsewhere) are used to simulate the magnetization in a 2d setup at finite temperatures in state-of-the-art model and with methods are well-known and established.

Response 1: We thank the Reviewer for reading our manuscript and for his/her thoughtful comments and suggestions. We have carefully analysed all points raised by the Reviewer and provide scientific data through new results and analysis that further support our findings. We have expanded the results presented in our work to address all comments raised by the Reviewer. A point by point response is included in the following, with the location referring to the revised manuscript.

The movies are beautiful to look at but the analysis is disappointing (see below for a few examples). Finally, the authors summarize that merons and antimerons might be created stochastically when cooling down from large temperatures but that they will decay via pair annihilation. This conclusion seems to me somewhat trivial as of course defects of all kind might be stabilized when quenching a system and the pair creation and annihilation of merons (or vortices, skyrmions, etc) is also well-known (including a Nobel prize?).

Response 2: *We thank the Reviewer for his/her kind words regarding our movies. As mentioned in the abstract, the amount of layered materials that have shown the presence of merons or anti-merons in their magnetic structure is very scarce in condensed matter (e.g. Refs. 1, 5-8). This has limited substantially the study of the most fundamental spin-quasiparticle (e.g. merons, antimerons), which does not appear intrinsically in every system (Ref. 27). Chiral materials are more susceptible to stabilize merons and antimerons which is not the case of monolayer CrCl₃ (Ref. 27). Indeed, the vortex formation in crystalline materials is very rare since it is normally destroyed by long-range interactions resulting in trivial spin textures such as magnetic domains (Ref. 2). In this aspect our manuscript shows that CrCl₃ developed such behaviour given its unicity through the interplay between different quantities as discussed in **Responses 2-5** for Reviewer 1. Therefore, our findings provide a new platform to the study of merons in 2D materials.*

The lifetime, mentioned in the title, is not discussed in this manuscript.

Response 3: *The main idea behind the use of lifetime in the title is to show the creation, evolution and annihilation of meron and antimeron topological spin textures in CrCl₃. However, in its present form it could be misleading for the reader.*

Modifications: *We have modified the title to “Properties and dynamics of meron topological spin textures in the two-dimensional magnet CrCl₃”.*

I therefore do not see any scientific advance in this report and have concerns about the interpretation of some findings. I can not recommend it for publication -- in particular not in a journal with a high impact as Nature Communications.

Response 4: *As we discussed below, we have expanded the discussions and results in our manuscript to address all Reviewer's comments. We provided additional dataset that remarkably support our findings. We hope that these modifications can satisfactorily convince the Reviewer regarding the scientific advances of our manuscript.*

In particular concerning the analysis of the simulations, I have a number of questions and serious concerns.

(1) The topological charge in Eq.4 is only quantized to integers if the integration area is a closed surface. Alternatively, the boundary of the area can be “trivial” in some sense, e.g., polarized. None is the case here, therefore the charge should depend on the integration area (which the authors do not specify). Moreover, the charge cannot be quantized to $\pm 1/2$ but only to integers. So this analysis is very questionable. Also the explanation of the formula itself is very questionable, e.g., \hat{s} is of course the magnetic moment and not only its out-of-plane component, as claimed by the authors.

Response 5: *The Pontryagin index or the topological number N (Eq. 4) is one of the cornerstones of the analysis of topological objects including not only skyrmions or merons*

(Refs. 25, 29), but in the broad field of topology (see Abdalia et al. *Non-perturbative Methods in Two-Dimensional Quantum theory*). Eq. 4 has been used in several studies on merons and skyrmions (Refs. 2-8) with high accuracy on the explanation of the magnetic properties of several systems. In this aspect we are confident that our analysis and interpretation are accurate and valid.

Indeed, values of N can have non-integer numbers, ($N=\pm 1/2$) such as for vortex structures (merons, antimerons). Since at asymptotically large radii, S^z vanishes to minimize the system total energy whereas at the vortex core S^z tends to a maximum value to remove any singularity in the gradient energy. This is a result widely accepted in the literature used by different groups (Refs. 26-29) and a milestone in the definition of meron topological spin textures. Without considering a meron with non-integer topological number ($\pm 1/2$) we cannot even define it (Ref. 26-29).

Moreover, as commented in **Response 7** for Reviewer 1, the total charge of a meron or an anti-meron is $\pm e/2$, where e is the electrical charge. In the manuscript, we did not state that the charge is quantized but rather assume this value because it may depend of the state filling ν . (Ref. 29). The fact of merons carry fractional charge $\pm e/2$ can be deduced through a Berry phase argument as discussed in **Response 7** for Reviewer 1. **Figure R6** shows an extract from Ref. 29 regarding the fractional charge of a meron that clearly supports our arguments.

Moreover, we have not stated in any part of the manuscript that we are using only the out-of-plane component of the magnetic moment to evaluate N via Eq. 4. We considered all three components (S^x , S^y , S^z) of the magnetisation \mathbf{S} . In fact, we carefully studied the convergence of the integral in Eq. 4 regarding the area considered to extract the topological number N . The optimized area to compute N is 20 nm X 20 nm.

Modifications: We have included further details in the evaluation of the integral in Eq. 4 with a clear statement of the area converged in the simulations (pages 6-7, lines 111-122). Furthermore, additional references supporting our analysis were also included.

(2) The authors continue in line 115 that topology should imply stability which is not true, in particular when the antiparticle is present in the system.

Response 6: As discussed in **Response 6** for Reviewer 1, the non-trivial spin textures should not collapse or merge as it was observed initially in Figures 1-2. We are aware of that for a meron to be created in a system an antimeron should also be stabilized. Both spin quasiparticles are stable for the time of roughly 15-20 ns before the collisions take place as explained in the manuscript. Indeed, in comparison with other systems (Ref. 5, 30, 31), this lifetime is at least two orders of magnitude higher which certainly is an advantage for experimental validation.

Modifications: We have changed the sentences related to the stability of the spin quasiparticles (page 7, lines 123-125).

(3) The authors notice the spiraling orbit during collision, lines 123 and following, which is also well-known for vortices and can be described simple analytical arguments that use collective coordinates. In fact, a lot of findings that are reported in this paper are not new but references and a maybe deeper analysis are missing. For example, the long-range interaction between defects in this setting would be interesting to draw conclusions on the lifetime and also the role of the biharmonic interaction would be interesting to know.

Response 7: In the context of 2D magnetic vdW materials, which is a recent field in magnetism, our findings are unique, novel and stabilize a new paradigm on the search for non-trivial topological spin textures which have not been observed in CrCl_3 . To the best of our knowledge no other 2D vdW magnets have shown evidences of merons and antimerons in

their magnetic structure. The possibility to stabilize meron topological spin textures in a non-chiral material opens new endeavours on the exploration of rich physical phenomena involving the interactions between vortex and antivortex without Dzyaloshinskii Moriya interaction.

Furthermore, as commented in **Response 3** for Reviewer 1, our results drive the search for other 2D vdW magnets with similar magnetic environment which might allow the creation of non-trivial topological spin-textures. The mechanism discussed in our manuscript provides a general framework for further investigations.

In our calculations, we have not considered any biharmonic driving force (e.g. $f(t) = f_1 \sin(m\omega t) + f_2 \sin(n\omega t + \varphi)$) applied to CrCl_3 , but rather the time evolution of the Landau-Lifshitz-Gilbert (LLG) equation spanned over long times. In this framework, we can explore how thermal fluctuations can affect the properties of the magnetic vortex and antivortex found in our system.

Modifications: We have included additional details on the lifetime of meron and antimerons, in CrCl_3 (page 7, lines 123-130), and also an estimation from our dataset.

(4) The authors compare their simulations with dipolar fields to simulations without dipolar fields but otherwise the same parameters. That obviously does not yield inplane magnetizations as the very strong inplane anisotropy by the dipolar interaction is missing. In studies of thin magnetic layers or monolayers, this is usually corrected by approximating the dipolar interaction in monolayers by an effective local inplane anisotropy which would stabilize inplane magnetic textures here. Therefore, this comparison is invalid and should be improved. I suspect that the dipolar interactions can be entirely replaced by a local easy plane anisotropy, otherwise distortions of the antivortex should be visible because of its

Figure R7: Snapshot of a spin dynamics of CrCl_3 at 8 K without considering dipolar interactions at any direction but setting an easy-plane (XY) for the magnetic anisotropy ($120.7 \mu\text{eV}$).

nonzero magnetic charges (which are also not analyzed in this manuscript).

Response 8: As commented in **Responses 2-4** for Reviewer 1, the main driving force to stabilize the merons and antimerons in CrCl_3 is the interplay between different exchange interactions, dipolar fields and magnetic anisotropy. We can not really assign just one quantity behind their formation. Our manuscript described carefully the physics behind this effect as we don't impose any restrictions on the spins and the simulations are undertaken self-consistently. That is, the atomic scale is described accurately through strongly correlated DFT+U approximations, and the micromagnetic part via LLG equation techniques.

Nevertheless, we understand that other approaches to describe the formation of vortices in thin-films assumed an effective local in-

plane anisotropy to simplify the analysis of the magnetic interactions (Refs. 42-43). We explicitly checked the Reviewer's suggestion by removing the dipole-dipole interactions from our simulations and setting an easy-plane anisotropy in CrCl₃ (**Figure R7**). The spin textures formed at this artificial easy-plane looked more chaotic than those computed without such restriction being more complex to assign any clear feature or to determine a topological number N .

Modifications: We have included **Figure R7** as a new Supplementary Figure S11, with further discussions in the manuscript (pages 11-12, lines 209-214).

(5) Concerning the presentation of the results, I don't understand why all spin components use the same color code (which is confusing) while sometimes different color codes are used for the same quantity (see Fig.S3). Do the authors consider variations of the spin length (this is not explicitly mentioned)? If not, a color scheme that covers the 2-sphere could be much more easy to understand for the reader. I also don't understand how the simulation cell looks like. What is the lattice underlying all those figures? This is not clear to me from the text in the supplement.

Response 9: Figure S3 used a different colour code due to the large magnetic fields utilized. For field beyond 150 mT (bottom panels in red-blue scale in Fig. S3) the panels will be pitch-dark without allowing the observations of any features. On the new scale, few features are still observed at 175 mT and 200 mT.

Regarding the spin length, there are no variations of the spin length in any part of the manuscript or in the simulations.

Regarding the underlying lattice, at the present scale (400 nm X 400 nm) no features are able to be observed. The focus of the figures 1-5 is to show the spin features of the non-trivial textures which are much larger than the lattice (few Å's). The simulation lattice including the Mn atoms can be visualized in Figure S5.

Modifications: We have included few more sentences clearly explaining why a second colour scale was utilized in the caption of Figure S3.

Reviewer 3:

The manuscript by Augustin et al. describes the formation and annihilation of meron-type topological spin structure in CrCl₃, a currently fashionable two-dimensional magnet. The work is based on an extensive simulation formalism involving Monte Carlo and LLG spin dynamics calculations. The predictions are made for a monolayer sheet of CrCl₃.

The results presented are very interesting and shed some light on the micromagnetism in v.d.W. materials. The CrCl₃ sheet is shown to host merons and antimerons, which spontaneously form below the theoretically calculated Curie temperature (around 19 K). The dynamic behaviour of the topological structures is analysed as a function of temperature and magnetic field, showing that they eventually annihilate resulting in a homogeneous magnetisation. In my view the paper is concise, but still leaves some open questions, which should be answered before publication can be considered. The questions are listed below.

Response 1: We thank the Reviewer for his/her kind words regarding our manuscript. We have carefully reviewed our manuscript and included additional results that expand the scope of our findings. These modifications provided further support to our work and consideration for publication in Nature Comm.

1. The spin Hamiltonian which forms the basis for the calculations does not contain - apart from a small single ion anisotropy - any further anisotropies or terms related to spin-orbit coupling. For a free-standing CrCl₃ monolayer this approach may be justified. In any realistic experiment the CrCl₃ sheet will have to be supported by a substrate, which may introduce additional interactions. To what extent will an additional exchange or spin-orbit interaction with the substrate affect the results?

Response 2: The spin Hamiltonian in Eq.1 includes spin-orbit coupling, apart from D_i (single ion anisotropy), in the calculation of the bilinear exchange J_{ij} (J_{xx} , J_{yy} , J_{zz}), anisotropic exchange λ_{ij} (λ_{xx} , λ_{yy} , λ_{zz}) and biquadratic exchange K_{ij} (K_{xx} , K_{yy} , K_{zz}). All the details are showed in Ref. 23.

Regarding additional interactions, we have checked whether additional spin-orbit effects might change our results. For the former, we included Dzyaloshinskii-Moriya interactions (DMI) in the spin Hamiltonian in Eq. 4 via $H_{DMI} = A \sum_{ij} (S_i \times S_j)$. We have done simulations with $A=0.05$ meV, 0.1 meV to provide a range of values for any critical behaviour. We assumed that an additional DMI on CrCl₃ would occur perpendicular to the surface due to a underneath substrate. **Figure R8** shows DMI has no effect on the formation of merons and antimerons in CrCl₃ which still occurred during the cooling process.

We have also computed the interlayer exchange interactions between CrCl₃ layers resulting in ~ 0.70 meV for first nearest neighbours and 0.002 meV for second-nearest neighbours. In comparison with the intralayer exchange considered in our simulations (Ref. 23), these magnitudes are two orders of magnitude smaller, which suggested very small perturbations on the spin-textures observed in CrCl₃. Moreover, we have recently become aware of experimental results of the synthesis of monolayer CrCl₃ on graphene/6H-SiC(0001) using molecular beam epitaxy (MBE) as shown in Ref. 49. In such setup the interactions between layer and substrate were observed to be very weak not affecting the magnetic structure of the sheet. This indicates that a system composed of CrCl₃/graphene/6H-SiC(0001) would be an ideal platform for the observation of non-trivial topological spin textures.

Modifications: We have included further discussions in Eq. 1 regarding the calculation of all terms including spin-orbit coupling (page 4, lines 65-67) and highlight the effect of spin-orbit in the simulations via DMI (Supplementary Figure S14).

Figure R8: a-b, Snapshots of a spin dynamics of CrCl₃ at 0 K taking into account Dzyaloshinskii-Moriya interaction (A) with $A=0.05$ meV and $A=0.1$ meV, respectively. The out-of-plane component of the magnetization S^z is utilized for following the evolution of the spin-textures.

2. The movies mapping the cooling down process seem to indicate that even at temperatures far below T_c (temperature window 5 - 20K) thermal fluctuations suppress the meron/antimeron formation. What is the reason for this strong influence of fluctuations? One reason mentioned is the small single-ion anisotropy. Would an increase of the SIA stabilise the situation? Could this be a prediction for other material systems beyond CrCl_3 to show similar phenomena?

Response 3: *Regarding the first question, the meta-stability of the magnetic domains in CrCl_3 and consequent continuous evolution of magnetization (Figure 1, and Supplementary Movie S1) is one of the main factors for the strong thermal fluctuations as our new simulations unveiled. **Figure R9** shows that even after CrCl_3 achieved 0 K in the zero-field cooling there is a consistent evolution of the magnetization in all its components (S^x , S^y , S^z) as a function of time. Although any thermal contribution to the magnetic domains will be zero at this limit (0 K), the spins would still evolve to stabilize the ground-state via the minimization of other contributions of the total energy, e.g. exchange, anisotropy, dipole interactions. In this regard, the small single-ion anisotropy (SIA) is a contributing factor.*

*Regarding the second question, an increment of SIA will not change the behaviour of the thermal fluctuations which will still be present (**Figure R10**). However, there is no formation of merons and antimerons. **Figure R10** shows that as we used a SIA slightly larger ($36 \mu\text{eV}$) than the one utilized in the simulations ($12.6 \mu\text{eV}$) all spins turned out-of-plane throughout the surface even considering dipolar interactions. This result indicates that the fluctuations are intrinsic to CrCl_3 .*

Modifications: *We have included **Figures R9** and **R10** in the main text and SI, respectively, with following discussions (pages 13-14, lines 236-257).*

Figure R9: *a*, Snapshot of a spin dynamics of monolayer CrCl₃ obtained through zero-field cooling after 2.00 ns and reaching 0 K. The magnetization perpendicular to the surface (S^z) is displayed showing the formation of merons and antimerons (small dots). Bright (dark) areas correspond to $S^z = \pm 1$, respectively in the colour scale. A path (dashed line) of 200 nm is drawn to show the spatial variation of the magnetization at different times ($t \geq 2.20$ ns). **b-d**, Variations of the in-plane components of the magnetization (S^x , S^y) and S^z , respectively, along of the path shown in **a** within 2.20-3.80 ns after 0 K is reached. The inset in **d** shows a small area from **a** along the path with the formation of the merons and antimerons. The corresponding variation of S^z at different times at A, B and C is also showed.

Figure R10: Snapshot of a spin dynamics for CrCl_3 at 0 K using a single-ion anisotropy slightly larger ($36 \mu\text{eV}$) than the one utilized in the simulations ($12.6 \mu\text{eV}$). There is no formation of merons or antimerons with the spins turning out-of-plane throughout the surface.

3. The nucleation of merons and antimerons appears to happen only on domain boundaries. The distance between neighbouring topological structures varies a lot from event to event. Why are the domain walls preferred and what determines the distance between the merons?

Response 4: As discussed in the manuscript, the domain wall profile is the region where the competition between different exchange interactions resulted in zero magnetization along of S^x and S^y . In such regions, the out-of-plane magnetization (S^z) become full polarized due to the large dipolar interactions. This is directly related with the energy stability of the system since a full in-plane spin polarization along S^x and S^y would lead to a singularity of the exchange energy which is avoided as S^z becomes non-negligible (Ref. 34). During the formation of the domain walls, we have not observed any factor or physical ingredient that would help to determine the distance between the merons/antimerons as they appear spontaneously and randomly in the zero-field cooling. The distance between the merons/antimerons in this sense follows the spin dynamics of the system being stochastic as our calculations indicated.

Modifications: We have included additional comments regarding the preference of the merons at the domain walls and the distance between the different spin textures in the manuscript (pages 8-9, lines 148-167)

4. Do the domain patterns and topological features form at the same time during the cool down process or is there a time lag between them? This is difficult to tell from the movies.

Response 5: Our simulations indicate that the magnetic domains need to be formed slightly before the topological features take place at the domain wall (Supplementary Movie S6). There is a short time lag between them of roughly ~ 0.10 - 0.20 ns.

Modifications: We have included this new data in the manuscript (page 8, lines 144-148).

5. In their discussion of the vortex-antivortex annihilation process (p.7) which results in the emission of spin waves the authors cite Ref. 28. However, this reference deals with the switching of a vortex core. Hertel et al. (PRL 2006) showed that this process involves a complex intermediate state with two additional vortex cores of opposite polarity created, one of which annihilates with the original vortex core releasing spin waves. Why do such intermediate states not seem to play a role in the meron/antimeron annihilation processes?

Response 6: *The intermediate state in Hertel et al. (PRL 2006) is observed when vortex-antivortex-vortex textures (Figures 2-3 in Hertel et al.) are brought together in a cross-tie structure. This situation is very different to the one observed in our simulations. In our manuscript we studied in detail collisions involving a vortex-antivortex pair, which is a two-body problem in essence. In Hertel et al., their calculations involved a three-body problem collision involving two vortices and one antivortex. In such complex annihilation process, an intermediate state is formed since two spin-textures (e.g. antivortex and vortex) are initially merged or combined before interacting with a third one (e.g. vortex). This makes the appearance of singularities into the system more susceptible.*

6. The title of the paper reads “lifetime evolution”, but I don’t find a quantitative determination of the lifetime for the various parameter sets studied. The movies cover a range of 40 ns. However, it is difficult to discern, for example, whether or not the strength of the applied field affects the lifetime of the merons/antimerons or just the density during the formation. Information of the actual lifetimes as a function of temperature and magnetic field would be a critical information for experimentalists. Their primary question may be: what needs to be done to bring the lifetime of the topological structures into a regime compatible with an experimental approach to measure/image them?

Response 7: *The lifetime extracted from our calculations is within the range of 15-20 ns, which corresponds to the time interval since they are created, displaced following the domain walls and collapse through collisions. The formation of spin textures happened mainly below 5 K. External magnetic fields do not affect their lifetime, but it has an effect on the polarization of the core of the vortex and antivortex. Figure S3 and Supplementary Movie S3 show that at large fields just one type of spin polarization is observed following the field direction. This resulted in a different number of vortex and antivortex with different orientation of S^z which can be translated in the density of spin textures with $S^z=+1$ and $S^z=-1$.*

The lifetime extracted from our simulations (15-20 ns) is at least two-orders of magnitude higher than that measured in other compounds that hold merons and antimerons in their magnetic structure, such as in ferromagnetic iron layer (Ref. 30), kagome magnet (Ref. 31), and permalloy disks (Ref. 5). This is a clear advantage for their imaging and observation using magneto-optical techniques, e.g. Magneto-optic Kerr effect microscope (MOKE), Spin Transmission Electron microscope (Spin-TEM), Lorentz Transmission Electron microscope (LTEM), etc. Moreover, good quality samples that hold weak interaction with substrates (Ref. 49) and a resolution below 50 nm for a microscopy technique (e.g. MOKE or Spin-STEM) would be needed to image/measure the topological textures predicted in our manuscript.

Modifications: *We have included a new section at the end of the manuscript (page 14-15, lines 259-280) titled “Implications and Prospects” that provides several comments regarding the experimental realization, guidelines to find merons/antimerons in other 2D vdW materials, and implications of our results.*

Reviewers' Comments:

Reviewer #1:

Remarks to the Author:

The authors have done a good job (mostly) responding to my previous comments. In particular, I appreciate the work they have done to expand the manuscript on what the roles of different interactions are as well as comparisons with other 2D magnetic systems.

We can continue to quibble over "topologically protected" merons. The fact is that the integral over all 2D space (or a closed 2D manifold) cannot yield a Pontryagin index of $1/2$ (or corresponding Chern class). This is only obtained only by integrating over a region in space, as the authors have done. If space were infinite or closed, then you cannot have single merons but pairs of merons and antimerons. These are technical details but the field has become a bit sloppy when it comes to calling things "topologically protected".

Other than this detail (which is not essential but I would appreciate if the authors, and other in the field, were a little more careful), the revised manuscript is fine and I can recommend it for publication.

Reviewer #2:

Remarks to the Author:

I have checked again the manuscript and to my delight, some of my former criticism was considered.

However, my main concern still is that I do not see any significant scientific advance in this paper. I don't want to discuss the Monte Carlo simulations here because they seem to be irrelevant for the main content: the merons. The authors predict to temporarily stabilize merons when very rapidly cooling a single layer of CrCl₃. They don't write it explicitly where I would expect this information (around line 73 where they describe the temperature protocol), but in a caption of a figure, that the time scale for cooling from above T_c to 0K is around 2ns. That is of course far below the natural timescale of the vortex dynamics. Effectively, this system is not "rapidly cooled" but quenched and the dynamics are essentially the same as starting from a random state and just keeping it at 0K. As I said in my previous report and as also the other referees say, the temporary emergence of merons in a quenched system with strong inplane anisotropy due to dipolar interactions is not a surprise. This is what I get from a system with only dipolar interactions and exchange interaction when evolving a random state at 0K for 1ns, using the free software package MuMax3 (see attached figure) \diamond Merons. I guess the color code is self-explaining. And even though I do not have the adequate references at hand, merons are the natural defect to expect when quenching a system which shows a local formation of inplane magnetization patches. There seems to be a problem, though, with the new data that the authors show where they claim to never observe the formation of merons if they neglect the one or the other tiny extra interaction. But I also don't understand why they show dynamics "at 5K" if what they should compare to are 0K dynamics from the main text. This seems very strange to me.

Let me summarize: To me, the formation of merons in this setup (monolayer quenched easy plane magnet) is trivial. I don't see why it should be published in a high impact journal such as Nature Communications.

Reviewer #3:

None

NCOMMS-20-11745A

Response to Reviewers Comments:

Reviewer 1: The authors have done a good job (mostly) responding to my previous comments. In particular, I appreciate the work they have done to expand the manuscript on what the roles of different interactions are as well as comparisons with other 2D magnetic systems.

We can continue to quibble over "topologically protected" merons. The fact is that the integral over all 2D space (or a closed 2D manifold) cannot yield a Pontryagin index of $1/2$ (or corresponding Chern class). This is only obtained only by integrating over a region in space, as the authors have done. If space were infinite or closed, then you cannot have single merons but pairs of merons and antimerons. These are technical details but the field has become a bit sloppy when it comes to calling things "topologically protected".

Other than this detail (which is not essential but I would appreciate if the authors, and other in the field, were a little more careful), the revised manuscript is fine and I can recommend it for publication.

Response: *We thank the Reviewer for reading the updated version of the manuscript and for accepting it for publication. We really appreciated all comments, suggestions and analysis suggested by the Reviewer. We also agree with the Reviewer that some authors in the community have not been so careful when describing topological properties based on fundamental arguments. We believe that our manuscript will help to solidify the ideas suggested by the Reviewer to provide further progress and developments in the field.*

Reviewer 2: I have checked again the manuscript and to my delight, some of my former criticism was considered.

Response 1: *We thank the Reviewer for reading the updated version of the manuscript and to be delighted with the modifications. We have carefully addressed all additional comments raised by the Reviewer and modified the manuscript accordingly. A point-by-point revision is showed below, with the location referring to the revised manuscript.*

However, my main concern still is that I do not see any significant scientific advance in this paper. I don't want to discuss the Monte Carlo simulations here because they seem to be irrelevant for the main content: the merons.

Response 2: *As mentioned in our previous Response Letter, the methods developed in our manuscript allow that a large amount of materials being studied similarly as CrCl_3 in a systematic and accurate manner. Our approaches open new avenues for the exploration of non-trivial spin-textures in the timely field of 2D magnetic materials. Moreover, applications can also be envisioned throughout our findings which are scientific relevant for implementation, harvesting and understanding of novel physical phenomena in the atomic limit. Indeed, the Monte Carlo and LLG simulations bridge up the atomistic (few Å's limit) and macroscopic worlds (several nm's), without them it would be unlikely to observe the formation, evolution and annihilation of merons and anti-merons in CrCl_3 . In this case, the combination of *ab initio* and atomistic approximations is instrumental to the study of spin-textures in layered materials.*

The authors predict to temporarily stabilize merons when very rapidly cooling a single layer of CrCl₃. They don't write it explicitly where I would expect this information (around line 73 where they describe the temperature protocol), but in a caption of a figure, that the time scale for cooling from above T_c to 0K is around 2ns.

Response 3: *We have explicitly included this information (2 ns) in line 73 as missed by the Reviewer.*

That is of course far below the natural timescale of the vortex dynamics. Effectively, this system is not "rapidly cooled" but quenched and the dynamics are essentially the same as starting from a random state and just keeping it at 0K. As I said in my previous report and as also the other referees say, the temporary emergence of merons in a quenched system with strong inplane anisotropy due to dipolar interactions is not a surprise.

Response 4: *As we carefully described in our previous Response Letter, the fine interplay between various ingredients such as exchange, anisotropy and dipole-dipole interactions, play a major role in the creation of merons and anti-merons. It is not only one ingredient, e.g. in-plane magnetic anisotropy, that plays the key on the formation of non-trivial topological spin textures, but rather a cooperation of several factors. This is clearly stated in our manuscript and its Supporting Information. In addition, while we now know they exist as defects is perhaps not surprising after the presentation of the results in our paper, but details of their structure and their time evolution is an important effect that may have important consequences. The cooling is not instantaneous but as we clearly state the merons form in an intermediate state that for example can be induced with ultrafast laser pulses and may have applications in reservoir computing where the dynamic evolution of the system is important. There is also an important distinction between forming some uncharacterised multidomain state as the Reviewer's simulation shows, and fully characterising its dynamic formation, magnetic structure and time evolution as we present in our paper. This analysis alone represents a significant advance and development of understanding of the properties of complex non-uniform magnetic structures.*

This is what I get from a system with only dipolar interactions and exchange interaction when evolving a random state at 0K for 1ns, using the free software package MuMax3 (see attached figure)  Merons. I guess the color code is self-explaining. And even though I do not have the adequate references at hand, merons are the natural defect to expect when quenching a system which shows a local formation of inplane magnetization patches.

Response 5: *We thank the Reviewer very much for undertaking simulations on his/her own in order to reproduce some of the results included in the manuscript. We however need to be careful with the statement that the spin configurations observed in the Reviewer's simulations are conclusively merons. As described carefully in our manuscript, a thorough analysis on the topology of such textures (e.g. Pontryagin index) is necessary, including the orientation of the magnetic moments, and on the parameters used. Moreover, no references are included with the figure included by the Reviewer with a lack of essential detail to verify the scientific approach. Therefore, even though we appreciated that the Reviewer has dedicated time/resources to backup our statements, we stand in a position that without further analysis and a scientific sound and external peer-reviewed evaluation, the Reviewer's data is of limited validity. Indeed the universality of our findings to other systems is an interesting development and opens the possibility of observing similar effects in other materials.*

There seems to be a problem, though, with the new data that the authors show where they claim to never observe the formation of merons if they neglect the one or the other tiny extra interaction.

Response 6: *All the new data included in our manuscript and Supporting Information has been thoroughly reviewed and checked before submission. All the interactions have been carefully analysed and no issues have been found.*

But I also don't understand why they show dynamics "at 5K" if what they should compare to are 0K dynamics from the main text. This seems very strange to me.

Response 7: *The dynamics at 5 K and even below that show the appearance of thermal fluctuations (lines 128, 238) that plays an important role in the creation of the merons and anti-merons. Such fluctuations are still observed near 0 K (Figure 5) which drive the system to thermal instabilities responsible for the formation of monodomains at longer times. A thorough analysis has been included in the manuscript regarding these results.*

Let me summarize: To me, the formation of merons in this setup (monolayer quenched easy plane magnet) is trivial. I don't see why it should be published in a high impact journal such as Nature Communications.

Response 8: *As mentioned in Response 1, our manuscript sets a new ground on the exploration of non-trivial topological particles in 2D magnetic materials. Long-range magnetic order in 2D materials is new and in particular without the symmetry breaking necessary for the overcoming of the Mermin-Wagner theorem (no long-range magnetic order at finite T is possible without uniaxial anisotropy). Our results directly contradict this long assumed theory, and we show definitively that the magnetism is in fact much more interesting and exciting than expected. Our methods will allow the understanding of a large number of compounds that are either being isolated or chemically designed in different labs worldwide. Indeed, as we discussed page 14 in 'Implications and Prospects', an experimental realization of such systems is reachable, and we are confident that our predictions may be confirmed in the near future. Therefore, we believe that our results deserve publication in a high-profile journal as Nature Communications.*

Reviewers' Comments:

Reviewer #2:

Remarks to the Author:

I have read the authors' replies to my previous comments and find that they corrected their manuscript in only one particular point, namely the 2ns time scale in line 73.

Let me reply to the authors' replies one by one again:

(2) This manuscript does not contain any newly developed methods. The news seems to be that you might find metastable merons if you quench CrCl₃, based on the material parameters that were calculated in an earlier publication.

(3) Thank you for now including this information.

(4+5) I am still certain that the observed behavior is absolutely standard for a magnetic monolayer or thin film in which the uniaxial anisotropy is smaller than the dipolar interactions. In my last report I did not have any references at hand. But luckily the authors already provide them, e.g., Ref.5 (see Fig.4 in there).

The authors' reply #5 also shows that they seem to still not understand the controversy with Reviewer 1, as they claim that my merons would somehow be less topological than their own merons. None of them are truly topological and, in particular, none of them are "protected". To support my very simple simulations (which took me about 3 minutes to set up and simulate, it can be composed from the examples on the github website of MuMax3), I have given all the relevant parameters in my previous report: I only used an exchange interaction and dipolar interactions, starting from a random state, and relaxing it at 0K. With these parameters, the micromagnetic problem is uniquely defined and does not have any free parameters except the system size which is also somewhat arbitrary. And of course, there is an intermediate timescale where merons appear, as they naturally appear in a multidomain inplane polarized background. I am therefore very disappointed that the authors claim that my data would be "of limited validity", given that they know all the details.

Let me add, furthermore, that there are no well-defined domain walls in the manuscript. This is just an artefact of the color code and presentation scheme that was chosen by the authors and should either become apparent when adapting my color scheme (which is standard for magnetic simulations) or it could prove me wrong.

(6+7) Once again: The merons in the main text form around 5K and are mostly analyzed at 0K. Most of the (still very qualitative and unfortunately not quantitative) analyses are at 0K. Then the impact of changed material parameters should also be analyzed at 0K. According to the above literature and my own simulations, I still believe that merons will form upon quenching to 0K if the inplane anisotropy due to dipolar interactions dominates. If this is not the case, however, the authors should analyze why the simple and intuitive picture fails. At present, I still find the analysis insufficient and not suitable for Nat. Comm.

I also don't understand which thermal fluctuations the authors are referring to at 0K. The line-scans in Fig.5 don't have any meaning for me because the merons are mobile and not pinned to this line, so what is the meaning of this plot?

(8) There is no new kind of 2d long-ranged order being reported in this manuscript which deals uniquely with unstable quasiparticles the decay with time. Therefore, I also don't see why Mermin-Wagner should play a role here.

Concerning the experimental realization, the authors claim that Ref.5 would be a useful reference. But the Fresnel LTEM images that were taken in this reference only capture the final state after relaxation, not the state immediately after the femtosecond pulses. The acquisition timescale of LTEM (in Fresnel mode) is usually of the order of microseconds and not suitable for the observation of nanosecond excitations as reported here.

Let me summarize my criticism once again: This manuscript does not provide any insights beyond what is known, namely that a thin (or monolayer) magnet with a negligible uniaxial anisotropy shows short-lived merons when quenched from $T > T_c$ to $T = 0K$. The authors do not provide any quantitative analysis beyond this rather trivial result, which could have been a single sentence in their previous publication where they calculated the material parameters.

I acknowledge that this new material class currently attracts a lot of attention. Therefore, this information, even if trivial, might trigger new experiments and interest. But due to the poor analysis, I cannot recommend publication in Nature Communications with its high scientific standards.

Response to Reviewers Comments:

Reviewer 2: I have read the authors' replies to my previous comments and find that they corrected their manuscript in only one particular point, namely the 2ns time scale in line 73. Let me reply to the authors' replies one by one again:

(2) This manuscript does not contain any newly developed methods. The news seems to be that you might find metastable merons if you quench CrCl₃, based on the material parameters that were calculated in an earlier publication.

Response 1: *Our developed methods and analysis are the state-of-the-art not available in many groups worldwide. The small scale of merons in our simulations cannot be reproduced in micromagnetic calculations as their finite temperature ($T>0$) dynamics will also be wrong (see **Response 3** below). The micromagnetics also does not include more complex interactions (e.g. biquadratic), or describe potential competitions between them such as higher-order exchange and dipolar fields. Hence, there are large and important quantitative differences between our simulations and the oversimplified micromagnetic calculations suggested by the Reviewer.*

(3) Thank you for now including this information.

(4+5) I am still certain that the observed behavior is absolutely standard for a magnetic monolayer or thin film in which the uniaxial anisotropy is smaller than the dipolar interactions.

Response 2: *As we have carefully discussed in our previous Response Letters (1st, 2nd), the combination of several ingredients (e.g. exchange, dipolar interactions, on-site anisotropy) plays the key role in the stabilisation of merons in CrCl₃. We cannot really affirm that a magnetic monolayer or thin film will develop topological non-trivial spin textures as the Reviewer stated without citing any references or providing scientific evidences. This issue was already discussed in the first Response Letter (**Response 3 to Reviewer 1, Response 2 to Reviewer 2**).*

Nevertheless, we do know that CrCl₃ develops merons and anti-merons in its magnetic structure, but we are not aware of any other 2D magnet developing such effect. We stand in the position that arguments should be scientific verified via peer-review evaluation in order to be scientific relevant.

In my last report I did not have any references at hand. But luckily the authors already provide them, e.g., Ref.5 (see Fig.4 in there).

The authors' reply #5 also shows that they seem to still not understand the controversy with Reviewer 1, as they claim that my merons would somehow be less topological than their own merons. None of them are truly topological and, in particular, none of them are "protected".

Response 3: *We would like to point out that there wasn't any controversy with **Reviewer 1** regarding any aspect of his/her report. We carefully addressed all his/her comments in the updated version of the manuscript. Indeed, **Reviewer 1** happily recommended our manuscript for publication in the previous interaction similarly as **Reviewer 3**.*

We agree with the Reviewer that the essential phenomenon of the formation of magnetic topological structures (either transient domain structures, Skyrmions or merons) is a natural consequence of cooling (either quenching on the ps timescale, or slowly over the ns timescales we described). However, any concrete analysis of the nature of these structures is completely absent in Fig. 4 of Ref. 5. Indeed, merons or anti-merons are not mentioned at all

over the entire Ref. 5. Only the terms 'vortex' and 'anti-vortex' are used which from a more rigorous point of view are not completely precise. Because the Pontryagin index needs to be calculated explicitly and more analysis carried out similarly as those undertaken in our manuscript. However, this is a problem observed across the literature in general. This was also highlighted by **Reviewer 1** in his/her final comments before accepting our manuscript (Response letter 2).

Notwithstanding that the Reviewer has not provided any detail or reference of his/her simulation data, it is questionable the applicability of the micromagnetic formalism (as the one implemented in Mumax3 code used by the Reviewer) for quenching simulations. That is, the micromagnetic exchange formulation requires a small angle between neighbouring regions of magnetization, unlike the Heisenberg exchange in our present calculations using atomistic methods. In fact, one of us did a careful comparison between micromagnetic methods (Mumax3) and atomistic approaches (as the one used in our manuscript) in a recent study (Iacocca et al. Nature Communications (2019)10:1756, <https://www.nature.com/articles/s41467-019-09577-0>). The best comparison with the experimental results is always obtained with the atomistic methods (e.g. Figure 7 in Iacocca et al.). Moreover, the details of the complex interactions in CrCl₃ including long range and higher-order exchange interactions, atomic resolution of the dipole fields make the actual dynamics, stability and size of the merons unique to this material and its physical properties.

To support my very simple simulations (which took me about 3 minutes to set up and simulate, it can be composed from the examples on the github website of MuMax3), I have given all the relevant parameters in my previous report: I only used an exchange interaction and dipolar interactions, starting from a random state, and relaxing it at 0K.

Response 4: The Reviewer has never provided any relevant parameters or details of his/her simulations in his/her previous reports. As mentioned in our previous response, we don't consider the Reviewer's results as relevant for the discussions to our manuscript as no reference or peer-review process was evaluated on the Reviewer data.

Moreover, calculations performed in 'about 3 minutes' using tutorial 'examples' from a web server as the Reviewer mentioned do not provide the reliability and trustiness of a careful study over several months as that undertaken in our manuscript.

With these parameters, the micromagnetic problem is uniquely defined and does not have any free parameters except the system size which is also somewhat arbitrary. And of course, there is an intermediate timescale where merons appear, as they naturally appear in a multidomain inplane polarized background. I am therefore very disappointed that the authors claim that my data would be "of limited validity", given that they know all the details. Let me add, furthermore, that there are no well-defined domain walls in the manuscript. This is just an artefact of the color code and presentation scheme that was chosen by the authors and should either become apparent when adapting my color scheme (which is standard for magnetic simulations) or it could prove me wrong.

Response 5: As mentioned in Response 4, the Reviewer has never provided the details of his/her simulations in any interactions with us. We don't know how they were performed, what parameters were used, etc. His/her data was produced and used without being previously evaluated in a peer-review assessment. We identify this as a not good practice in the evaluation of our manuscript.

In addition, our manuscript presents a detailed study of the dynamics and time-evolution (several nanoseconds) of the merons and anti-merons. This is new and allows us to quantify the behaviour of the topological magnetic structures and their fascinating dynamics in a long temporal scale. On the colour scheme, we have applied a quite standard colour scheme to

distinguish the different vertical orientations of the merons and anti-merons, while the in-plane scheme mentioned by the Reviewer neglects this information, since until recently they have not been classified at all.

While the term "domain wall" is usually applied in simpler cases such as a 1D transition, they physically only represent the transition between correlated regions of magnetization, and so we disagree that domain walls are not present in our system. We have carefully studied how magnetic domains evolve throughout CrCl_3 at different temperatures, magnetic fields, parameters and parent compounds (Figure 1, Figures S1-S3, Figures S7-S15, Movie S1-S3, Movie S7-S9). Based on our data, it is undeniable the presence of magnetic domains in CrCl_3 . Indeed, recent experimental results (Ref. 50) suggested similar behaviour.

(6+7) Once again: The merons in the main text form around 5K and are mostly analyzed at 0K. Most of the (still very qualitative and unfortunately not quantitative) analyses are at 0K. Then the impact of changed material parameters should also be analyzed at 0K.

Response 6: *The merons are analysed at 0 K because their features and properties are observed much clearer than at 5 K where thermal fluctuations are large. This is explained at length in the manuscript where we provided a quantitative analysis on the main aspects of the merons, anti-merons and their dynamics. The parameters utilized in our simulations are those corresponding to CrCl_3 . If we change the parameters, we will artificially change the material. This discussion is not relevant to the main findings of the manuscript.*

According to the above literature and my own simulations, I still believe that merons will form upon quenching to 0K if the inplane anisotropy due to dipolar interactions dominates. If this is not the case, however, the authors should analyze why the simple and intuitive picture fails. At present, I still find the analysis insufficient and not suitable for Nat. Comm.

Response 7: *As we discussed above in Responses, 3, 4, 5, the Reviewer's simulations do not provide any scientific sound argument to reject our manuscript. They were neither evaluated externally in a peer-review assessment, nor directly provided any evidence of the Reviewer's argument. Indeed, the Reviewer has not provided any reference that support his/her claims. We sincerely believe that such practice used by the Reviewer to reject our manuscript should be prevented by the journal.*

I also don't understand which thermal fluctuations the authors are referring to at 0K. The line-scans in Fig.5 don't have any meaning for me because the merons are mobile and not pinned to this line, so what is the meaning of this plot?

Response 8: *The line scan in Figure 5 shows the time-evolution of the three components of the magnetization (S_x , S_y , S_z). We are not only showing the dynamics of the merons and anti-merons, but also how domain walls change once 0 K is achieved in the cooling. This is a new information to understand the meta-stability of the magnetic domains in CrCl_3 . This is discussed in detail in the manuscript.*

(8) There is no new kind of 2d long-ranged order being reported in this manuscript which deals uniquely with unstable quasiparticles the decay with time. Therefore, I also don't see why Mermin-Wagner should play a role here.

Response 9: *As discussed in the previous response letter (Response 8), we don't have any uniaxial anisotropy for CrCl_3 . This would be needed for magnetic order at finite T following the Mermin-Wagner theorem. Our results directly contradict this long-assumed theory, and we*

show definitively that the magnetism is in fact much more interesting and exciting than expected.

Concerning the experimental realization, the authors claim that Ref.5 would be a useful reference. But the Fresnel LTEM images that were taken in this reference only capture the final state after relaxation, not the state immediately after the femtosecond pulses. The acquisition timescale of LTEM (in Fresnel mode) is usually of the order of microseconds and not suitable for the observation of nanosecond excitations as reported here.

Response 10: *Ref. 5 shows an example of a system with vortex and anti-vortex in a femtosecond transient state. We have never stated in any part of the manuscript that the merons and anti-merons in CrCl₃ would behave similarly as the Reviewer is implying. The observation of merons in CrCl₃ would depend on several factors as explained in the Implications and Prospects. Moreover, the formation of merons and anti-merons in CrCl₃ happened during the cooling, whereas the vortex and anti-vortex in Ref. 5 are obtained through an in-situ laser excitation in permalloy (Py) disks. Even though both systems (Py and CrCl₃) may have similarities, they behave differently on their own.*

Let me summarize my criticism once again: This manuscript does not provide any insights beyond what is known, namely that a thin (or monolayer) magnet with a negligible uniaxial anisotropy shows short-lived merons when quenched from $T > T_c$ to $T = 0K$. The authors do not provide any quantitative analysis beyond this rather trivial result, which could have been a single sentence in their previous publication where they calculated the material parameters.

I acknowledge that this new material class currently attracts a lot of attention. Therefore, this information, even if trivial, might trigger new experiments and interest. But due to the poor analysis, I cannot recommend publication in Nature Communications with its high scientific standards.

Response 11: *As mentioned previously, the methods developed in our manuscript allow that a large amount of materials being studied similarly as CrCl₃ in a systematic and accurate manner. They quantitatively provide a framework to unveil new avenues for the exploration of non-trivial spin-textures in the timely field of 2D magnetic materials. All methods are included in the main text and SI, which will benefit the community for further implementations.*

The study of non-trivial spin-textures in magnetism is a field on its own with several fundamental questions, intrinsic problems, and applications (see Refs. 1, 23, 27, 31). We cannot summarize all the developments performed in our manuscript by using a “single sentence” as the Reviewer suggested. We agree with Reviewer that we should simply as much as possible, but an adequate representation is paramount for additional progress. Therefore, we believe that our results provide a leap in the field and will drive novel exciting findings deserving our manuscript publication in a prime journal as Nature Communications.

Reviewers' Comments:

Reviewer #2:

Remarks to the Author:

I have once again read the replies of the authors to my previous criticism. My previous concerns were, mainly, that the physics (formation of merons in a quenched film) are rather trivial and that the authors claim to have developed new methods which is not the case (or at least I don't understand which part of the methods is supposed to be new).

Unfortunately, when reading the authors' replies, I do not find any attempt to explain what precisely is new, neither for the physics part nor for the methods, or any other argument why this manuscript should be published in Nat. Commun. It is still unclear to me what the news are. Instead of getting a real answer, I only read meta-answers, e.g., "Our developed methods and analysis are the state-of-the-art not available in many groups worldwide". This statement might be correct, yet it does not add any information that might potentially be useful.

In addition, the analysis is absolutely scarce and does not add any insights beyond showing movies and pictures (and the trivial fact that merons and antimerons can annihilate, which is also just mentioned but not analyzed). Hence, I cannot imagine that the article is of any interest for the broader readership of Nat. Commun. The authors could at least have tried a simple Thiele analysis for the motion or to calculate a domain wall profile (which, I believe, does not make sense because there are no well-defined domains, see previous reports).

And, finally, I have expressed strong doubts about the correctness of the results: I have argued that the emergence of short-lived merons after quenching from $T > T_c$ to $T = 0$ is naturally expected if the effective inplane anisotropy is stronger than the effective uniaxial anisotropy which I even have supported by my own simulations. This intuitive result seems to not match the simulations of the authors. I have tried to explain in my previous reports where this discrepancy might come from (e.g., finite T vs $T = 0$) but the authors seem to not care about this apparent contradiction and keep claiming that my efforts should be disregarded as they are not peer-reviewed. I find this rather absurd as this is the peer-review process and one of the authors has experience with MuMax3, so it should have been an easy task to check my data if you don't believe it. Instead I am being accused of using "not good practice", which is again wrong and only proves that the authors have not understood the concept of scaling in continuum theories, which would make the results of the manuscript so much more universal and impactful. [1]

I have also mentioned in my previous report that the main reference for a potential experimental observation (Ref.5) was mis-interpreted by the authors and cannot be used to image the suggested metastable states. The images in Ref.5 are taken with LTEM which has a much longer acquisition time than the femtoseconds suggested by the authors. I kindly ask you to look at Fig. 4 in that reference again which explains that the observed states are metastable long-lived magnetic patterns. Only the exciting laser pulse is on the femtosecond scale, but not the imaging technique. However, the authors disregard my attempt to correct this mistake which I now call "not good practice".

This brings me to the conclusion that I cannot expect any further improvement of this manuscript. I cannot recommend to publish this manuscript in Nature Communications.

Footnote [1].

I wrote in the earlier report that I use the standard micromagnetic model in MuMax3 with only magnetic stiffness and demagnetization. I also claimed that this information is enough to reproduce my simulations. Here is why: The micromagnetic model is a continuum model. Therefore, you can rescale continuous quantities and make the model dimensionless. With only stiffness and demag, you can rescale the units of length and energy, and thereby absorb both the stiffness and the demag. After rescaling, there is no stiffness or demag parameter left that has to be specified. Moreover, the length scale is given by the magnetic length and the energy (or time) is given by a magnetic energy (or time). But as I wrote in the previous report: You just have to

choose a large enough system size and look at the correct time scale. Here is an example script for MuMax3 which gives you merons on the nanosecond time scale, using the parameters of the standard problem #4. I was looking for better parameters, but the manuscript does not contain them and only refers to another paper.

```
SetPBC(10, 10, 0)
SetGridsize(128, 128, 1)
SetCellsize(300e-9/128, 300e-9/128, 300e-9/128)
Msat = 800e3
Aex = 13e-12
alpha = 0.01
m = RandomMag()
autosave(m, 0.1e-9)
run(5e-9)
```

That said, the micromagnetic model is not a fixed model. You can always add new terms that you deem relevant for your problem, e.g., biquadratic exchange, etc. can and should be included. But nobody would consider this then a "new model".

Response to Reviewers Comments:

Reviewer #2 (Remarks to the Author): I have once again read the replies of the authors to my previous criticism. My previous concerns were, mainly, that the physics (formation of merons in a quenched film) are rather trivial and that the authors claim to have developed new methods which is not the case (or at least I don't understand which part of the methods is supposed to be new).

Unfortunately, when reading the authors' replies, I do not find any attempt to explain what precisely is new, neither for the physics part nor for the methods, or any other argument why this manuscript should be published in Nat. Commun. It is still unclear to me what the news are. Instead of getting a real answer, I only read meta-answers, e.g., "Our developed methods and analysis are the state-of-the-art not available in many groups worldwide". This statement might be correct, yet it does not add any information that might potentially be useful.

Response 1: *We thank the Reviewer for reading our manuscript and for his/her comments. As discussed thoroughly in our previous Response Letters 1, 2, 3, the methods developed and utilized in our manuscript set a new ground for the study of a variety of non-trivial spin textures in 2D vdW magnets. In brief, we use a discrete atomistic representation of the system that correctly accounts for: 1) the Heisenberg and non-Heisenberg contributions to the exchange interactions with all parameters calculated self-consistently via Hubbard-corrected Density Functional Theory, and 2) a direct dipole-dipole interaction between a truly 2D system. In fact, MuMax code uses the continuum approximation of the exchange energy ($E = A \theta^2$), valid only for small angle θ between spins (cells) - a fact that has been well known in the community for decades. This makes the representation of Bloch points (tightly wound magnetic structures) invalid, [C. Andreas et al. Phys. Rev. B 89, 134403, (2014)] and requires a fully atomistic description, which is the method we use in our manuscript. Furthermore, the representation of magnetic structures of high temperature states is similarly incorrect, and the evolution of these states in time is also incorrect using micromagnetics [E. Iacocca et al., Nature Communications 10, 1756 (2019)].*

In addition, the analysis is absolutely scarce and does not add any insights beyond showing movies and pictures (and the trivial fact that merons and antimerons can annihilate, which is also just mentioned but not analyzed). Hence, I cannot imagine that the article is of any interest for the broader readership of Nat. Commun. The authors could at least have tried a simple Thiele analysis for the motion or to calculate a domain wall profile (which, I believe, does not make sense because there are no well-defined domains, see previous reports). And, finally, I have expressed strong doubts about the correctness of the results: I have argued that the emergence of short-lived merons after quenching from $T > T_c$ to $T = 0$ is naturally expected if the effective inplane anisotropy is stronger than the effective uniaxial anisotropy which I even have supported by my own simulations. This intuitive result seems to not match the simulations of the authors. I have tried to explain in my previous reports where this discrepancy might come from (e.g., finite T vs $T = 0$) but the authors seem to not care about this apparent contradiction and keep claiming that my efforts should be disregarded as they are not peer-reviewed. I find this rather absurd as this is the peer-review process and one of the authors has experience with MuMax3, so it should have been an easy task to check my data if you don't believe it. Instead I am being accused of using "not good practice", which is again wrong and only proves that the authors have not understood the concept of scaling in continuum theories, which would make the results of the manuscript so much more universal and impactful. [1]

I have also mentioned in my previous report that the main reference for a potential experimental observation (Ref.5) was mis-interpreted by the authors and cannot be used to

image the suggested metastable states. The images in Ref.5 are taken with LTEM which has a much longer acquisition time than the femtoseconds suggested by the authors. I kindly ask you to look at Fig. 4 in that reference again which explains that the observed states are metastable long-lived magnetic patterns. Only the exciting laser pulse is on the femtosecond scale, but not the imaging technique. However, the authors disregard my attempt to correct this mistake which I now call “not good practice”.

This brings me to the conclusion that I cannot expect any further improvement of this manuscript. I cannot recommend to publish this manuscript in Nature Communications.

Response 2: *The analysis we have conducted is appropriate within the scope of our paper, showing the local spin structure of the merons/anti-merons and their dynamical properties. Despite the details included in the main text, a lengthy Supporting Information (SI) with 25 pages is included to our manuscript providing further information about our methods and analysis. Both the main and SI files encapsulate the essential physical behaviour of the structures allowing a general understanding of how they are formed, their annihilation properties and their metastability. Further work will consider their mean lifetime and thermal stability, but such studies are only recently feasible and will rely on similar recent studies of skyrmions published in the past months [B. Ding et al. Nano Letters 20, 868 (2020), M. Yang et al. Science Advances 6, 36, eabb5157 (2020)], yet their existence has been known for many years [A. Fert et al. Nature Nanotechnology 8, 152 (2013)].*

Moreover, we use a true 2D representation of the system of spins with direct dipole-dipole interactions and honeycomb symmetry which has a different form compared to thin films using the FFT representation of cubic cells used in MuMax. We therefore strongly disagree with the Reviewer's assertion that simple scaling allows one to map a complex system such as CrCl₃ onto a simplistic micromagnetic approximation. We do not doubt that the Reviewer's calculation in MuMax is straightforward and achievable. Nevertheless, we disagree that the results are physically meaningful for this 2D system with higher order magnetic interactions. Indeed, as mentioned in the previous Response Letters 2 and 3, the Reviewer's results have not been peer-reviewed which clearly limited their validity on the discussions.

In terms of physics, these states are physically accessible using ultrafast excitations such as lasers and current pulses which is also true for undiscovered 2D magnetic materials with higher ordering temperatures. Their applications are yet unknown but could provide a route to novel information processing and reservoir computing [D. Pinna et al. Phys. Rev. Appl. 14, 054020 (2020), D. Prychynenko et al. Phys Rev. Appl. 9, 014034 (2020), G. Bourianoff et al. AIP Advances 8, 055602 (2018)]. We thank the reviewer for pointing out the imaging technique timescale in Ref. 5, which we have reviewed our manuscript accordingly.